# Variance adaptation in navigational decision making

Ruben Gepner[1], Jason Wolk[1], Digvijay Shivaji Wadekar[1], Sophie Dvali[1], Marc Gershow[1,2,3]*

[1]Department of Physics, New York University, New York, United States; [2]Center for Neural Science, New York University, New York, United States; [3]Neuroscience Institute, New York University, New York, United States

**Abstract** Sensory systems relay information about the world to the brain, which enacts behaviors through motor outputs. To maximize information transmission, sensory systems discard redundant information through adaptation to the mean and variance of the environment. The behavioral consequences of sensory adaptation to environmental variance have been largely unexplored. Here, we study how larval fruit flies adapt sensory-motor computations underlying navigation to changes in the variance of visual and olfactory inputs. We show that variance adaptation can be characterized by rescaling of the sensory input and that for both visual and olfactory inputs, the temporal dynamics of adaptation are consistent with optimal variance estimation. In multisensory contexts, larvae adapt independently to variance in each sense, and portions of the navigational pathway encoding mixed odor and light signals are also capable of variance adaptation. Our results suggest multiplication as a mechanism for odor-light integration.
DOI: https://doi.org/10.7554/eLife.37945.001

*For correspondence:
mhg4@nyu.edu

Competing interests: The authors declare that no competing interests exist.

## Introduction

The world is not fixed but instead varies dramatically on multiple time scales, for example from cloudy to sunny, day to night, or summer to winter; and contexts: from sun to shade, or burrowing to walking. Nervous systems must respond appropriately in varied environments while obeying biophysical constraints and minimizing energy expenditure (*Niven and Laughlin, 2008*). Sensory systems use strategies at all levels of organization, from sub-cellular to network-level structures, to detect and transmit relevant information about the world to the rest of the brain, often operating near the limits imposed by their physical properties (*Bialek, 1987*; *Niven and Laughlin, 2008*).

One strategy to maximize the conveyance of sensory information is to reduce redundancy by filtering out predictable portions of sensory inputs (*Barlow, 1961*; *Attneave, 1954*; *Srinivasan et al., 1982*). Early evidence of efficiency in the way sensory neurons encode environmental information was found in the large monopolar cells of the blowfly retina, whose responses to intensity changes were found to be optimally matched to the range of stimuli encountered in the fly's natural environment (*Laughlin, 1981*). If neurons' responses are indeed tuned to a specific environmental distribution of inputs, then when environmental conditions change, the neurons' coding must adapt to the new environment's statistics to maintain an efficient representation of the world (*Tkačik and Bialek, 2016*; *Maravall, 2013*).

A rich field of study has explored adaptation of neural coding to the statistics, particularly the mean and variance, of sensory stimuli (*Wark et al., 2007*). Variance adaptation has been measured across diverse organisms in visual, auditory, olfactory, and mechanosensory systems and in deeper brain regions (*Brenner et al., 2000*; *Fairhall et al., 2001*; *Kvale and Schreiner, 2004*; *Dean et al., 2005*; *Nagel and Doupe, 2006*; *Dahmen et al., 2010*; *Maravall et al., 2007*; *De Baene et al., 2007*; *Wen et al., 2009*; *Gorur-Shandilya et al., 2017*; *Clemens et al., 2018*; *Smirnakis et al.,*

*1997*; *Gollisch and Meister, 2010*; *Clifford et al., 2007*; *Liu et al., 2016*; *Kim and Rieke, 2001*) pointing to a potentially universal function implemented by a broad range of mechanisms.

Classic experiments in blowfly H1 neurons showed that variance adaptation in the firing rate maximized information transmission (*Brenner et al., 2000*; *Fairhall et al., 2001*). Because the role of these neurons is to transmit information, we can say this adaptation is 'optimal,' in a well-defined theoretical sense. It is less clear how to test for optimality in the adaptation of behaviors to changes in environmental variance.

If the brain enacts behavior based on rescaled sensory input, one might expect that these behaviors should also adapt to stimulus variance. In rhesus monkeys performing an eye pursuit task (*Liu et al., 2016*), adaptation to variance was observed in both the firing rates of MT neurons, maximizing information transmission, and in the ultimate eye movements, minimizing tracking errors, showing that efficiency in neural coding has observable effects on behavioral responses.

On the other hand, animals choose behaviors to achieve an end goal. For general sensory-driven behaviors, this goal, for example flying in a straight line or moving toward a potential food source, does not require maximizing the transmission of information about the stimulus to an observer of the behavior. In these cases, it might be adaptive for the brain to restore some or all information about environmental variance before choosing which behaviors to express.

To explore the role of variance adaptation in a general sensory information processing task, we studied how *Drosophila* larvae adapt their navigational decisions to variance in visual and olfactory stimuli. Navigation requires the larva to transform sensory information into motor output decisions in order to move to more favorable locations, and the larva's navigational strategies have been characterized for a variety of sensory modalities (*Gomez-Marin and Louis, 2012*; *Gershow et al., 2012*; *Gomez-Marin and Louis, 2012*; *Gomez-Marin et al., 2011*; *Kane et al., 2013*; *Lahiri et al., 2011*; *Louis et al., 2008*; *Luo et al., 2010*; *Sawin et al., 1994*). *Drosophila* larvae move in a series of relatively straight runs interspersed with reorienting turns. A key element of the larva's navigational strategy is the decision to turn, that is to cease forward movement and execute a reorientation maneuver to select a new direction for the next run. This decision can be modeled as the output of a Linear-Nonlinear-Poisson (LNP) cascade and characterized through reverse correlation (*Gepner et al., 2015*; *Hernandez-Nunez et al., 2015*; *Aljadeff et al., 2016*; *Parnas et al., 2013*; *Schwartz et al., 2006*; *Chichilnisky, 2001*).

In this work, we investigate how the navigational decision to turn adapts to the variance of sensory input by characterizing how changes in stimulus variance lead to changes in LNP model parameters. We show that larvae adapt to the variance of visual stimuli and of olfactory inputs processed by a range of olfactory-receptor neurons. We find that adaptation can be characterized as a rescaling of the input by a factor that changes with the variance of the stimulus. We show that the timescales of turn-rate adaptation are asymmetric with respect to the direction of variance switches and are consistent with optimal estimation of environmental variance (*Wark et al., 2009*; *DeWeese and Zador, 1998*). We find that in a multi-sensory context, adaptation is implemented independently on visual and olfactory inputs and also in the pathway that processes combined odor and light signals. We propose that multisensory integration may result from multiplication of signals from odor and light channels.

## Results

As a model for how organisms adapt their behavioral responses to changes in environmental variance, we explored visual and olfactory decision making by larval *Drosophila*. In previous work, we modeled the larva's decision to turn (stopping a run in order to pick a new heading direction) as the output of a Linear-Nonlinear-Poisson (LNP) cascade (*Figure 1a*). This model took as its input the derivative of stimulus intensity, and we used reverse-correlation to find the LNP model parameters (*Figure 1b*).

In this work, we extended the analysis to ask how larvae adapt their responses to changes in the variance of the input stimuli. We again provided visual stimuli through blue LED illumination and fictive olfactory stimuli through red LED activation of CsChrimson expressed in sensory neurons. The illumination was spatially uniform and temporally fluctuating, with derivatives randomly drawn from normal distributions. The variance of these normal distributions dictated the amplitude of temporal fluctuations. We changed the variance of these distributions (*Figure 1c*), and studied the resulting

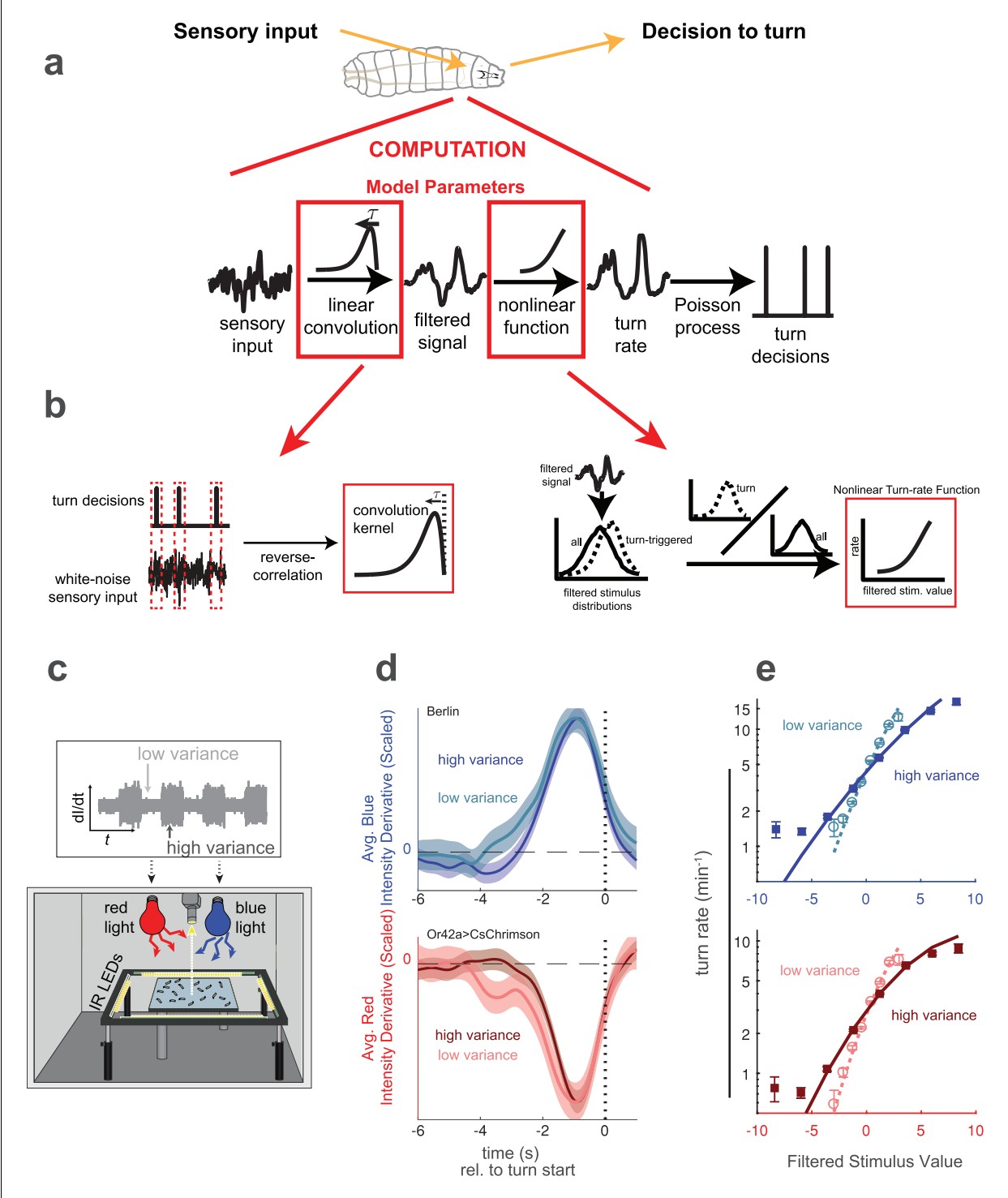

**Figure 1.** Reverse correlation analysis of variance adaptation. (**a**) Linear-Nonlinear-Poisson (LNP) model of the decision to turn. Sensory input is processed by a linear filter to produce an intermediate signal. The rate at which the larva initiates turns is a nonlinear function of this signal. Turns are initiated stochastically according to a Poisson process with this underlying turn rate. (**b**) Reverse correlation to determine LNP model parameters. The kernel of the linear filter is proportional to the 'turn-triggered-average', the average input preceding turns. The rate function is calculated by comparing the turn-conditioned filtered inputs to the entire input ensemble. (**c**) Experimental setup to test for behavioral adaptation to variance. We presented fluctuating light intensities - with randomly-picked derivatives - to groups of larvae crawling on an agar surface, and recorded their trajectories using IR lights and camera. Blue light provided a visual stimulus and red light activation of CsChrimson labeled neurons provided a fictive olfactory stimulus. The derivative of the light intensity was the input to our LNP model, and the variance of this input periodically changed between low and high values.
*Figure 1 continued on next page*

*Figure 1 continued*

(**d**) Turn-triggered averages measured at low and high variances for visual (Berlin, blue light) and fictive attractive odor (Or42a*CsChrimson*, red light) stimuli. To ease comparison of the temporal structure, the averages are scaled to have the same peak value. (**e**) Nonlinear rate functions measured at low and high variances. A single convolution kernel was calculated for the entire experiment; the turn rate as a function of the filtered input was calculated separately for low and high variance. Note logarithmic y-axis. *Number of experiments, animals in **Table 1***.

DOI: https://doi.org/10.7554/eLife.37945.002

The following figure supplements are available for figure 1:

**Figure supplement 1.** Overview of analysis steps.

DOI: https://doi.org/10.7554/eLife.37945.003

**Figure supplement 2.** Rate functions following single filter vs variance-specific filters.

DOI: https://doi.org/10.7554/eLife.37945.004

changes in the behavioral response as measured by changes in the LN model parameters (*Figure 1d,e*).

There is a subtle difference between stimulus and model input. For instance, in work on adaptation in blowfly H1 (*Brenner et al., 2000*; *Fairhall et al., 2001*), the stimulus was light reflected from a pattern of black and white stripes, while the model input was the horizontal velocity of this pattern. The experimenters then measured adaptation to the variance of this input.

In our experiments, the stimulus was projected light whose intensity varied in a Brownian random walk. The model input was the time-derivative of the stimulus intensity. This has two important advantages. First, it provides a mean zero input that is uncorrelated on all time scales, simplifying analysis and interpretation. Second, when we change the variance of the input, we change the statistics of how quickly the light level changes, but we do not change the mean or variance of the light intensity itself, eliminating potential confounds due to adaptation to overall light intensity.

## Larvae adapt their turn-rate to the variance in visual and olfactory sensory inputs

We first asked whether larvae adapt their behavior to changes in the variance of sensory input and if so, how. In one set of experiments, we presented wild-type larvae with blue light whose intensity derivatives were randomly drawn from normal distributions with low or high variance. In another set of experiments, we used a fictive olfactory stimulus generated by red light activation of CsChrimson expressed in Or42a sensory neurons.

In both sets of experiments, the environment switched between low and high variance ($\sigma^2_{high} = 9\sigma^2_{low}$) every 60 s. We expected that following a switch in variance, larvae would require some time to adapt their behaviors. We therefore discarded the first 15 s (this amount of time is justified later) following each transition and analyzed the larvae's responses separately in low and high variance contexts.

In the LNP framework, adaptation to variance could take two forms - a change in the shape of the linear filter kernel (*Kim and Rieke, 2001*; *Baccus and Meister, 2002*), representing a change in the temporal dynamics of the response, and/or a change in the shape of the nonlinear function, representing a change in the strength of the response to a particular pattern of sensory input. Both forms of adaptation have been reported in sensory neurons (*Wark et al., 2007*; *Maravall, 2013*). In adult *Drosophila*, ORN firing rates adapt to variance of both natural and optogenetic inputs through a change in the nonlinearity (*Gorur-Shandilya et al., 2017*).

We first examined the shape of the filter kernel. To find the kernel, we calculated the 'turn-triggered average' (*Gepner et al., 2015*), the average input preceding a turn, at both low and high variance. Given an uncorrelated input and a large number of turns, the kernel is proportional to the turn-triggered average (*Chichilnisky, 2001*), but the constant of proportionality is unknown and can be absorbed into the nonlinear stage. We therefore scaled both the low and high variance turn-triggered averages to have the same peak value. We found that for both visual and fictive olfactory stimuli, the scaled turn-triggered averages were nearly identical at low and high variance (*Figure 1d*), although at low variance, the averages had a slightly longer 'shoulder' 3–4 s prior to the eventual turn.

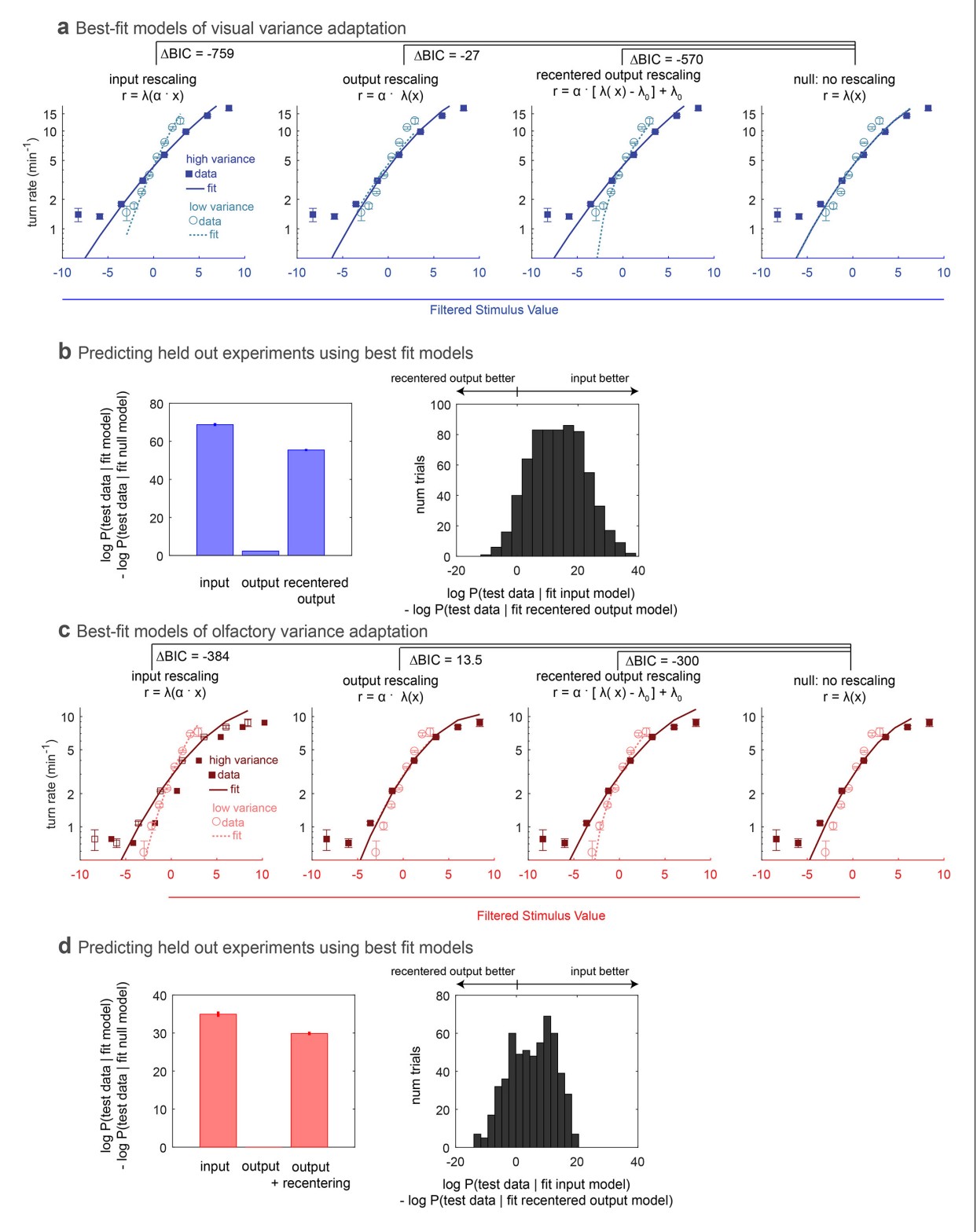

**Figure 2.** Comparing rescaling models of variance adaptation. (**a**) Best fit rate functions to visual (Berlin with blue light stimulus) data of *Figure 1*, for various functional forms of rescaling, and for a null model with no rescaling. $\Delta BIC$ indicates difference in Bayes Information Criterion between the adapting model and the null model. A negative value indicates the adapting model is preferred, and a difference of more than 10 is considered to be highly significant. (**b,c**) Ability of models to predict held-out data. In repeated 'trials,' each model was fit to a subset (14/17 experiments) of the data. The fit model was applied to calculate the likelihood of the data in the held out sets. Because test data sets varied in size, the log likelihood of each

*Figure 2 continued on next page*

Figure 2 continued

data set was normalized to the mean length of all test data sets. (b) Increase in log-likelihood of test data when predicted with rescaling models over the predictions of a null model without rescaling. Bars show mean increase and error bars the standard error of the mean. (c) Histogram of improvement of input rescaling model predictions over recentered output rescaling model predictions for each 'trial.' Δ log-likelihood > 4.6 corresponds to $p<0.01$. (d–f) Same as (a–c), but for fictive olfactory (Or42a>csChrimson with red light stimulus) data of panel 1. *Number of experiments, animals in* **Table 1**

DOI: https://doi.org/10.7554/eLife.37945.005

As the kernel shape did not strongly depend on the input variance, we first used the turn-triggered average for all data to estimate a single kernel for both low and high variance. We scaled this kernel so that the filter output had variance one in the low-variance condition. We then calculated the rate function at low and high variance to determine whether the magnitude of the larva's response to stimulus changes adapted to environmental variance (*Figure 1—figure supplement 1*). We found that for both visual and fictive olfactory stimuli the rate function (*Figure 1e*) was steeper at low variance than high variance, indicating a more sensitive response to the same input in the low variance context.

We then asked whether it might be more appropriate to use separate filters for low and high variance stimuli. We separately convolved the high and low variance stimuli with filters derived from the high and low variance turn-triggered averages (*Figure 1d*) and again calculated nonlinear rate functions at high (*Figure 1—figure supplement 2a,c*) and low (*Figure 1—figure supplement 2b,d*) variance. These rate functions were nearly identical to those we found using a single shared kernel. Thus, the different slopes of the rate functions at high and low variance (*Figure 1e*) are not due to mismatches in temporal dynamics between the filter derived from the pooled data and the variance-specific filters.

As it had little effect on the extracted rate functions and greatly simplified analysis, we used a single filter derived from pooled data in the remainder of our work.

## Larvae adapt to variance by rescaling the input

A larger response to the same input in a low variance context indicates variance adaptation. Several functional forms of adaptation have been proposed and measured. Rescaling the sensory input (*input rescaling*) by its standard deviation maximizes information transmission and has been observed in blowfly H1 neurons (*Brenner et al., 2000*) and in salamander retina (*Kim and Rieke, 2001*; *Kim and Rieke, 2003*), equivalently characterized as a change in amplitude of the linear filter kernel. On the other hand, many biophysically plausible models of variance adaptation (*Ozuysal and Baccus, 2012*) function by rescaling the *output* of a nonlinear function or by the summation of saturating nonlinear functions of correlated input (*Nemenman, 2012*; *Nemenman, 2010*; *Borst et al., 2005*).

We asked whether the adaptation we observed in the nonlinear rate function could be better described as an input or output rescaling. We used maximum likelihood estimation to find the best fit model of each form for the entire visual or olfactory data set and measured the improvement compared to the best-fit model without rescaling (*Figure 2a,d*). We found that an input rescaling far better described the adaptation than an output rescaling, a straightforward consequence of the high and low variance rate functions having the same non-zero output at zero stimulus input. We therefore also considered whether recentering followed by rescaling of the output could describe the adaptation, but again found that input rescaling better described the adaptation.

Next we asked whether the models differed in their abilities to predict held out test data. We fit each model to a subset of 14 out of 17 experiments and found the likelihood of the data in the held out three experiments given the fit model. For each permutation of fit and test data, we calculated the improvement in the log likelihood of the test data compared to the predictions of a null model without rescaling. As with the fits to the entire data set, we found that the input rescaling model was better at predicting held out data than either output rescaling model, in the aggregate (*Figure 2b, e*) and on a trial-by-trial basis (*Figure 2c,f*).

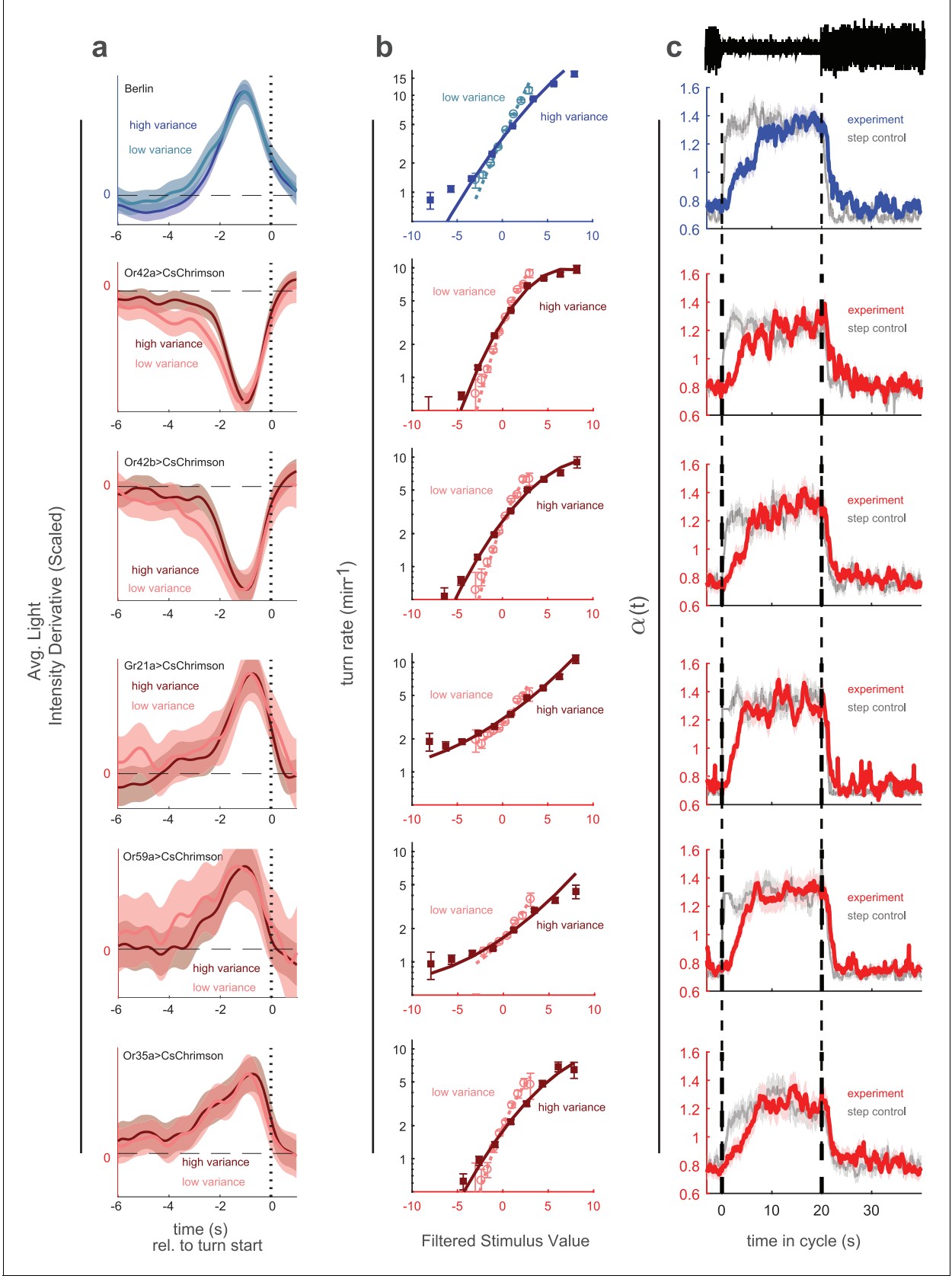

**Figure 3.** Variance adaptation and temporal dynamics of input rescaling. Larvae were exposed to alternating 20 s periods of high and low variance intensity derivative white noise. For Berlin, the stimulus was visual (blue light). For all other genotypes, the stimulus was optogenetic activation of the indicated receptor neuron type via CsChrimson and red illumination. (a) Turn-triggered averages measured at low and high variances. To ease comparison of the temporal structure, the averages are scaled to have the same peak value. (b) Nonlinear rate functions measured at low and high variances. A single convolution kernel was calculated for the entire experiment; the turn rate as a function of the filtered stimulus was calculated separately for low and high variance. Note logarithmic y-axis. (c) Scaling factor ($\alpha(t)$) vs. time since switch to low variance, averaged over many cycles. Schematic at top indicates low and high variance conditions. Colored lines (experiment) are maximum likelihood fits to data. Gray lines (step control) are maximum likelihood fits to data generated by a model in which the input rescaling changes instantly when the variance changes. $\alpha$ was normalized so that the average over the entire experiment was 1. *Number of experiments, animals in Table 1*

DOI: https://doi.org/10.7554/eLife.37945.006

## Asymmetric rates of variance adaptation to a variety of sensory inputs

Having found that larvae adapt to the variance of visual stimuli and of optogenetically induced activity in the Or42a receptor neuron, we next explored the timescales and generality of the adaptation. We expected that following a change in environmental variance, larvae would require some time to adapt their responses, a process that could be described by adding time dependence to the nonlinear rate function. Since we found that adaptation was best described as a rescaling of the input to a nonlinear function, we modeled the time-variation of the rate function as a time-varying input rescaling

$$r(t,x) \equiv \lambda(\alpha(t)x) \tag{1}$$
$$\lambda(x) = \lambda_0 \exp(bx + cx^2) \tag{2}$$

where $x$ is the output of the linear filter stage and $\lambda_0, b, c$ are constants to be determined.

To find the time scales of turn-rate adaptation to abrupt changes in sensory input variance, we decreased the period of the variance switching experiments to 40 s to increase the number of transitions. To test the generality of adaptation, we explored the visual response, activation of neurons that produce attractive responses (Or42a, Or42b), and activation of neurons that produce aversive responses (Or59a, Or35a, Gr21a). We found that in all cases, the linear filters were nearly the same

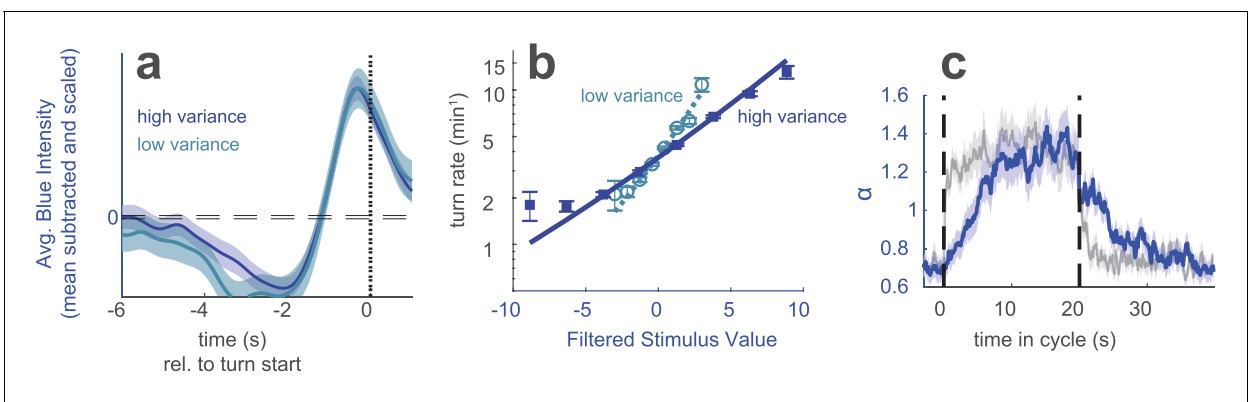

**Figure 4.** Variance adaptation to a stimulus with uncorrelated values. Berlin wild type animals were exposed to blue light. Every 0.25 s, the intensity of the light was chosen from a random normal distribution with fixed mean and low or high variance. (a) Turn-triggered average deviation from mean intensity, scaled to have same peak value at high and low variance. Analogous to *Figure 1d*. (b) Nonlinear rate functions measured at low and high variances. A single convolution kernel was calculated for the entire experiment; the turn rate as a function of the filtered input was calculated separately for low and high variance. Note logarithmic y-axis. Analogous to *Figure 1e*. (c) Scaling factor ($\alpha(t)$) vs. time since switch to low variance, averaged over many cycles. Colored lines (experiment) are maximum likelihood fits to data. Gray lines (step control) are maximum likelihood fits to data generated by a model in which the input rescaling changes instantly when the variance changes. $\alpha$ was normalized so that the average over the entire experiment was 1. Analogous to *Figure 3c*. *Number of experiments, animals in Table 1*.

DOI: https://doi.org/10.7554/eLife.37945.007

The following figure supplement is available for figure 4:

**Figure supplement 1.** Comparison of stimulii with uncorrelated random derivatives and uncorrelated random values.

DOI: https://doi.org/10.7554/eLife.37945.008

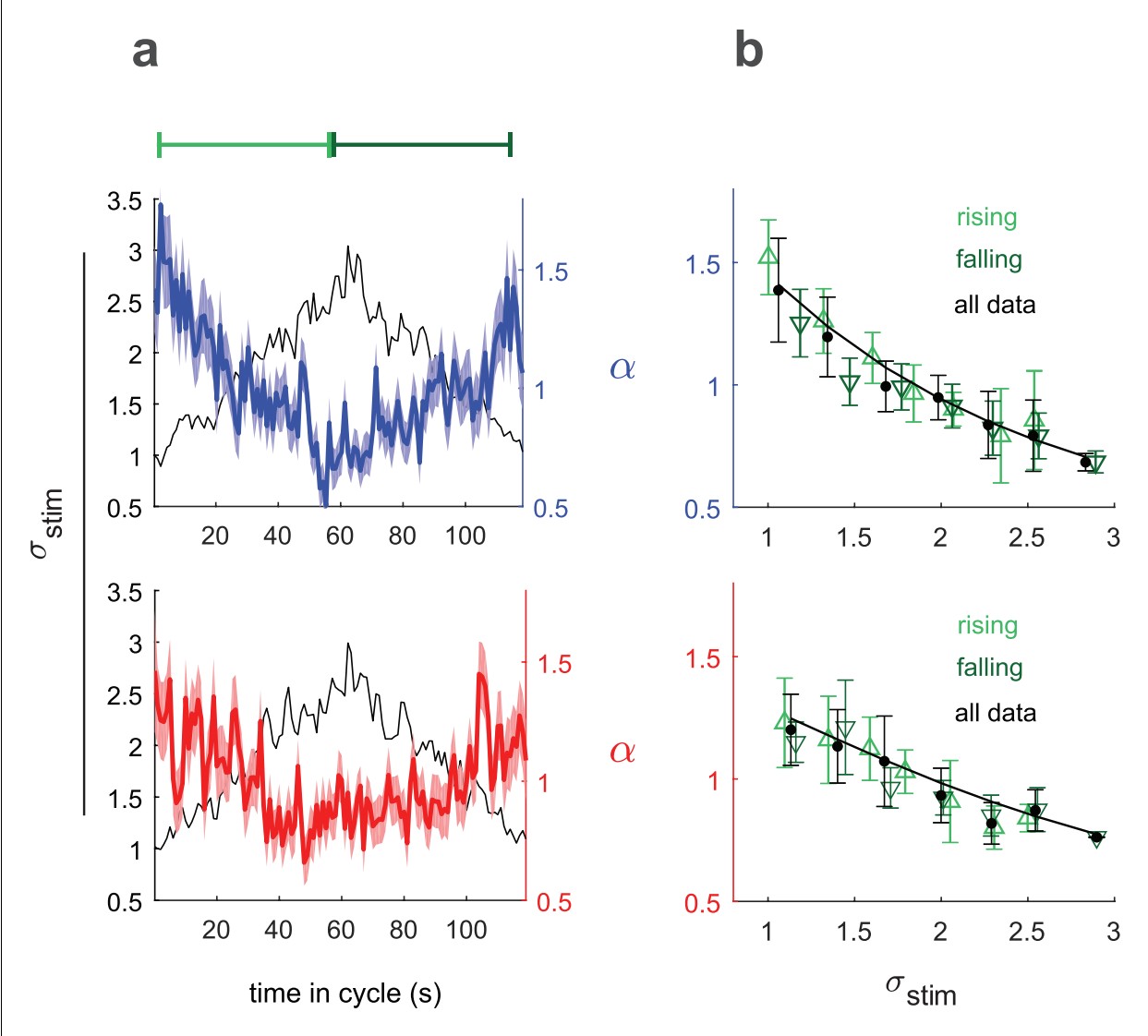

**Figure 5.** Input rescaling as a function of variance. Larvae were exposed to intensity derivative white noise whose standard deviation steadily increased and decreased in a 120 s period triangle wave. Top row: visual stimulus - blue light was presented to Berlin wild type. Bottom row: fictive attractive odor stimulus - red light activated CsChrimon expressed in Or42a receptor neurons. (a) Rescaling factors ($\alpha$) and stimulus standard deviation ($\sigma$) vs. time since variance started increasing. (b) Rescaling factor ($\alpha$) vs. stimulus standard deviation, as calculated from data in (a). Rising, falling indicate $\alpha$ vs. $\sigma$ calculated using only data when variance was increasing or decreasing respectively. Solid black line is a fit to *Equation 3* using all data. *Number of experiments, animals in Table 1*

DOI: https://doi.org/10.7554/eLife.37945.009

at high and low variances and that the slope of the nonlinear rate function adapted to variance (*Figure 3a,b*).

We then estimated the time-varying input rescaling $\alpha(t)$ (*Figure 3c*). We found that for all inputs this scaling factor decreased suddenly following a step increase in variance but increased more gradually following a step decrease. This is consistent with the response of an optimal variance estimator (*DeWeese and Zador, 1998*).

We asked whether the temporal dynamics in our estimate of $\alpha(t)$ arose from the larva's behavior or from the estimation process itself. To control for the latter possibility, for each set of experiments, we developed a control model in which $\alpha_{control}$ switched instantly along with the variance while all other parameters of the model were unchanged. We used this control model to generate a fictional set of turn responses to the same input stimulus, matching the number of experiments and larvae to

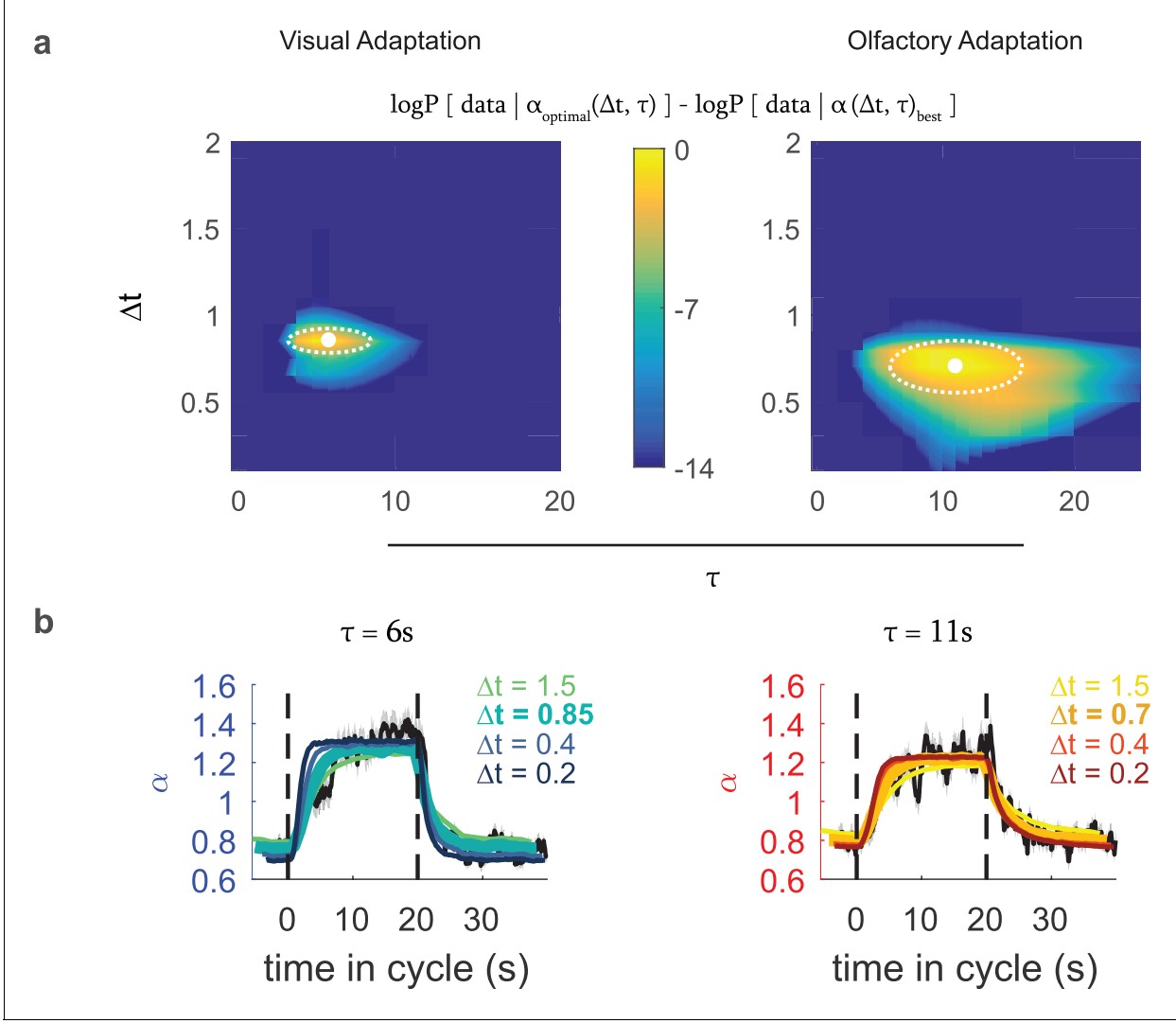

**Figure 6.** Optimal variance estimator predicts input rescaling. We generated an estimate of the input variance using a Bayes estimator that sampled the stimulus at an interval of $\Delta t$, with a prior $\tau$ that represented the expected correlation time of environmental variance, which we converted to an input rescaling parameter using the relation found in *Figure 5*. We found the static rate function parameters that in combination with this rescaling parameter best predicted the actual experimental data. We repeated this process for various combinations of $\Delta t$ and $\tau$ to find the one that best modeled the experimental data (a) Difference from the optimal log likelihood of data given model for various choices for $\Delta t$, $\tau$, for the visual and fictive olfactory (Or42a>CsChrimson) variance switching experiments of *Figure 3*. (b) Measured (black line) and predicted (colored lines) input rescaling vs. time for variance switching experiments, for the best fit value of $\tau$ and varying values of $\Delta t$. The thickest colored line is the prediction of the best-fit estimator. *Number of experiments, animals in Table 1*

DOI: https://doi.org/10.7554/eLife.37945.010

the original data set. We passed this fictive data set through our estimator and recovered $\alpha_{control}(t)$ (*Figure 3c*, gray line). We found that, the estimator always reported a sharp and symmetric transition in $\alpha_{control}$. Therefore the observed slower adaptation of $\alpha$ to variance decreases does not result from the estimation process.

## Variance adaptation under different stimulus statistics

In these experiments, we chose a stimulus whose *derivatives* at all time points were uncorrelated random gaussian variables. The trace of light intensity vs. time was therefore a Brownian random walk with reflecting boundary conditions, and changing the variance of the derivatives was equivalent to changing the diffusion constant (*Figure 4—figure supplement 1a,b*). An advantage of this

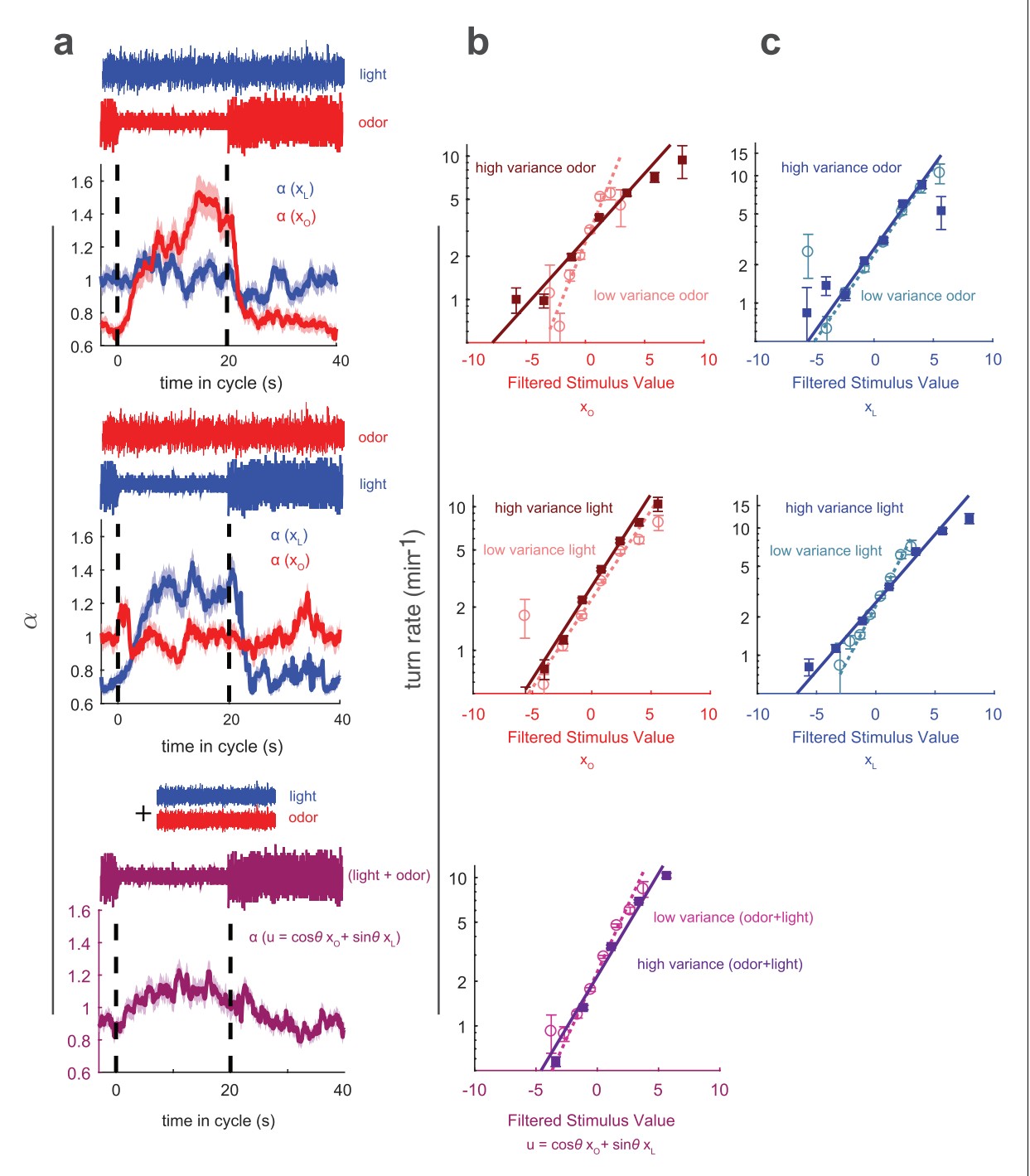

**Figure 7.** Multisensory variance adaptation and temporal dynamics of input rescaling. Or42a>CsChrimson larvae were exposed to both visual (dim blue light) and fictive olfactory (red light) stimuli with random intensity derivatives. *Top row*: visual input had constant variance, while olfactory input alternated between high and low variance every 20 s. *Middle row*: Olfactory input had constant variance and visual input alternated between high and low variance. *Bottom row*: Both olfactory and visual inputs had constant variance, but alternated between correlated and anti-correlated every 20 s. (**a**) Estimated input rescaling vs. time since variance switched to low, averaged over many cycles, for light and odor (top two rows) or combinations of light and odor (bottom row). (**b**) Turn rate vs. output of odor filter at high and low variance of the switching stimulus (top two rows) and vs. a linear combination of light and odor (bottom row). (**c**) Turn rate vs. output of light filter at high and low variance of the switching stimulus (top two rows). Note logarithmic y-axis in b,c. *Number of experiments, animals in* **Table 1**
DOI: https://doi.org/10.7554/eLife.37945.011

stimulus is that the light levels themselves are uniformly distributed over the entire range, for both low and high variance conditions (*Figure 4—figure supplement 1c*), so that adaptation to variance cannot be explained, for instance, as due to saturation of channels or receptors at high light intensities.

It is also possible to perform behavioral reverse correlation using a stimulus with uncorrelated random values (*Hernandez-Nunez et al., 2015*). We wondered whether we would also observe variance adaptation using such a stimulus. We exposed Berlin wild type larvae to a blue light stimulus with random intensity updated every 0.25 s. The mean of the intensities was constant, but the variance switched periodically ($\sigma^2_{high} = 9\sigma^2_{low}$). When the variance of the intensities switched so did the variance of the derivatives (*Figure 4—figure supplement 1g-i,k*). While the intensities were uncorrelated on all time scales (*Figure 4—figure supplement 1j*), subsequent changes in intensity were strongly anti-correlated (*Figure 4—figure supplement 1i*).

We first considered a stimulus whose variance changed every 60 s. We analyzed the responses as we did for the analogous visual experiments of *Figure 1*. The linear filter, found as the mean

**Table 1.** Number of experiments, animals, turns.
# experiments - number of 20 min duration experiments; a different noise input was used for each experiment within a group; # animals - estimate of total number of individual larvae surveyed in each set; animal-hours - total amount of animal-time analyzed; # turns total number of turns observed

| Figure | Genotype | # expts | # animals | animal-hours | # turns |
|---|---|---|---|---|---|
| *Figure 1,Figure 2* | Berlin | 17 | 811 | 219.3 | 48711 |
| | Or42a>CsChrimson | 17 | 743 | 201.1 | 33822 |
| *Figure 3* | Berlin | 30 | 1087 | 302.4 | 55776 |
| | Or42a>CsChrimson | 19 | 838 | 247.5 | 44806 |
| | Or42b>CsChrimson | 15 | 600 | 163.3 | 23826 |
| | Or59a>CsChrimson | 15 | 723 | 177.6 | 16418 |
| | Or35a>CsChrimson | 11 | 430 | 121.1 | 13149 |
| | Gr21a>CsChrimson | 19 | 786 | 230.1 | 36121 |
| *Figure 4a,b* | Berlin | 16 | 483 | 133 | 22741 |
| *Figure 4c* | Berlin | 18 | 699 | 189 | 28226 |
| *Figure 5* | Berlin | 10 | 372 | 102.5 | 18397 |
| | Or42a>CsChrimson | 13 | 553 | 147.2 | 21897 |
| *Figure 7* | Or42a>CsChrimson | | | | |
| Odor switches variance, visual constant | | 6 | 165 | 46.1 | 6772 |
| Odor constant variance, visual switches | | 10 | 391 | 105.4 | 15210 |
| Correlation switches | | 16 | 616 | 156.8 | 22142 |
| *Figure 9a* | Gr21a>CsChrimson | | | | |
| High Variance $CO_2$ | | 6 | 242 | 66.2 | 11286 |
| Low Variance $CO_2$ | | 6 | 242 | 62.4 | 10534 |
| *Figure 9a* | Or42a>CsChrimson | | | | |
| High Variance $CO_2$ | | 7 | 286 | 80.5 | 12976 |
| Low Variance $CO_2$ | | 7 | 276 | 77.9 | 12096 |
| *Figure 9b* | Gr21a>CsChrimson | | | | |
| High Variance $CO_2$ | | 3 | 121 | 33.2 | 7289 |
| Low Variance $CO_2$ | | 3 | 116 | 32.7 | 7023 |
| *Figure 9b* | Or42a>CsChrimson | | | | |
| High Variance $CO_2$ | | 2 | 94 | 25 | 5174 |
| Low Variance $CO_2$ | | 2 | 82 | 22.4 | 4571 |

DOI: https://doi.org/10.7554/eLife.37945.016

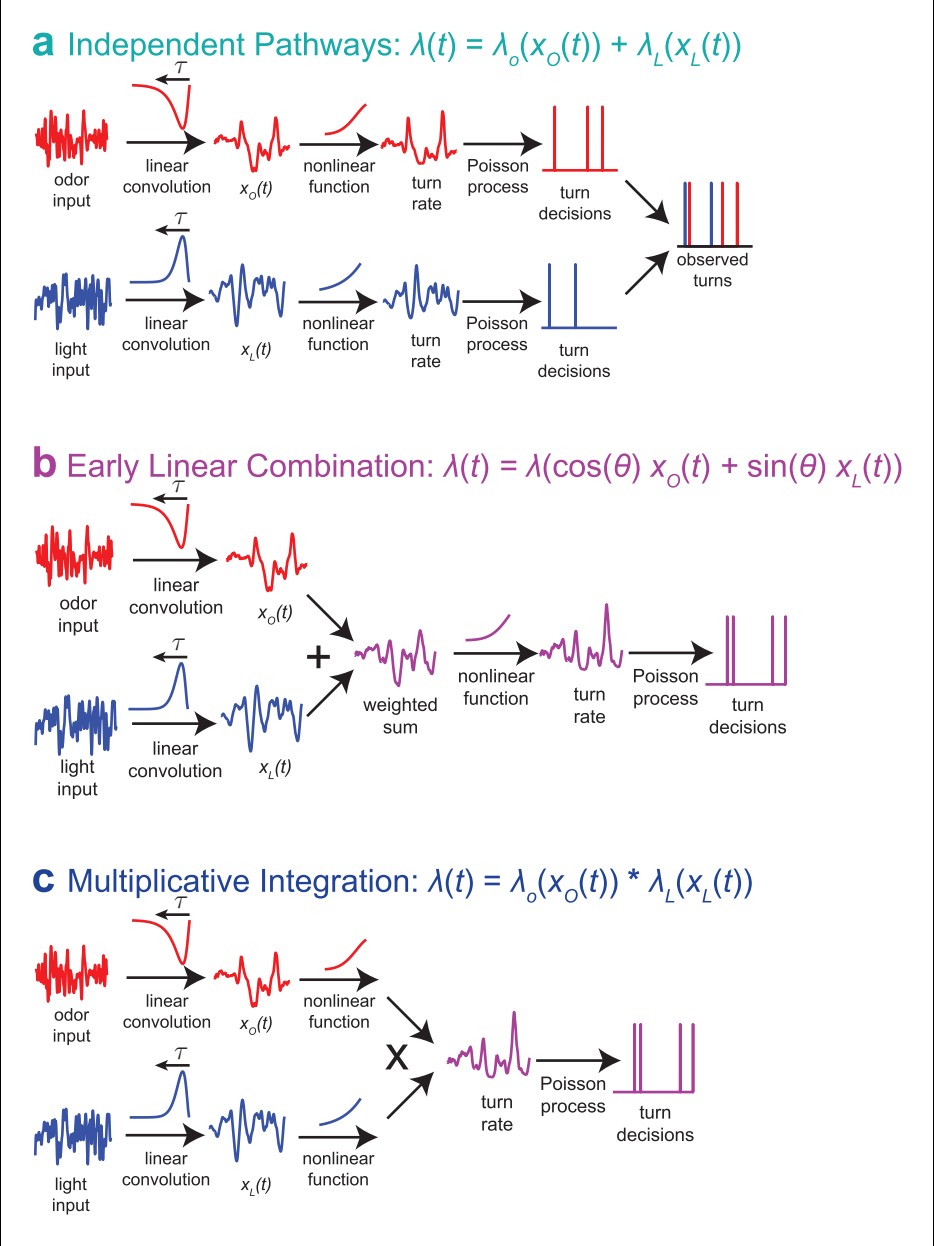

**Figure 8.** Multisensory combination models. (**a**) Independped pathways: two independent LNP models transform odor and light stimuli into decisions to turn; these turn decisions are combined by an OR operation at a late stage. This model is inconsistent with data from multisensory white noise experiments. (**b**) Early linear combination: Filtered odor and light signals are combined via linear summation prior to nonlinear transformation of the combined signal. This model is difficult to reconcile with adaptation to variance on a single-sense basis. (**c**) Multiplicative integration: Separate LN models for odor and light are combined via multiplication of the nonlinear function outputs. This model allows for unisensory variance adaptation via rescaling of a rectifying nonlinearity and is consistent with data from multisensory white noise experiments. Adapted from *Gepner et al. (2015)*.
DOI: https://doi.org/10.7554/eLife.37945.012

The following figure supplements are available for figure 8:

**Figure supplement 1.** Reanalysis of white noise experiments.
DOI: https://doi.org/10.7554/eLife.37945.013

**Figure supplement 2.** Reanalysis of multisensory step experiments.
DOI: https://doi.org/10.7554/eLife.37945.014

subtracted and scaled turn-triggered average, encoded the same temporal dynamics at low and high variance (*Figure 4a*). The nonlinear rate function was steeper in the low variance condition (*Figure 4—figure supplement 1b*) indicating variance adaptation. We then examined the temporal dynamics of adaptation using a stimulus whose variance switched every 20 s (in analogy to *Figure 3*) and found a faster adaptation to an increase in variance than a decrease (*Figure 4c*). These results all agreed with those of our experiments using uncorrelated stimulus derivatives.

## Larvae adapt to variance through a simple rule

Although the temporal dynamics of the input rescaling are consistent with an optimal estimate of the variance, it is not straightforward to test whether the observed $\alpha(t)$ truly reflects an optimal estimate. When the spike rate of a neuron adapts to environmental variance, there is a sound theoretical reason to believe it does so to maximize information transmission (*Brenner et al., 2000*; *Tkačik and Bialek, 2016*) and therefore that $\alpha \propto 1/\sigma$. However, for behavior, it is not clear if there is a theoretically optimal relation between $\alpha$ and $\sigma^2$.

We therefore sought to experimentally determine the relation chosen by the larva between input rescaling and observed variance. To do this, we slowly varied the stimulus variance in time in a triangular pattern and continuously monitored the scaling parameter $\alpha(t)$ (*Figure 5a*). If the variance changed slowly enough, the larva would be constantly adapted and the input rescaling $\alpha(t)$ could be taken to be a function of the variance $\sigma^2(t)$ at that time point only.

We related $\alpha(t)$, the time-varying rescaling parameter averaged over many cycles (colored lines, *Figure 5a*) to $\sigma(t)$, the cycle averaged stimulus standard deviation (black line, *Figure 5a*) to calculate $\alpha(\sigma)$ (*Figure 5b*). We estimated $\alpha(\sigma)$ using the full data set and separately for periods of increasing and decreasing variance. If the larvae were not continuously adapted to the variance, we would expect to see different estimates of $\alpha$ during the rising and falling phases, due to hysteresis. We found the same scaling vs. variance curve for the rising and falling phases, indicating that our measured $\alpha(\sigma)$ represented the larva's adapted rescaling law.

To analytically describe the larva's rescaling rule, we began with the input rescaling that maximizes information rescaling: $\alpha \propto 1/\sigma$. To better fit the data, we considered the possibility that the total variance might be due to both sensory input and other intrinsic noise sources: $\sigma^2_{total} = \sigma^2 + \sigma^2_0$, where $\sigma^2$ is the variance we introduce through the sensory input and $\sigma^2_0$, the intrinsic noise, is a fit parameter.

The solid line in *Figure 5b* shows the best fit to the data of this rescaling model

$$\alpha(\sigma) \propto \frac{1}{\sqrt{\sigma^2 + \sigma^2_0}} \tag{3}$$

This model can be interpreted as an attempt at 'optimal' rescaling in the presence of intrinsic noise $\sigma^2_0$ and/or as an adaptation to prevent large behavioral responses to minute changes in an otherwise static environment.

## Adaptation is consistent with an optimal variance estimate

Now that we could relate $\alpha(t)$, the measured input rescaling, to the larva's internal estimate of variance, we asked whether the larva's estimate of variance made the best use of sensory input. We constructed an optimal Bayes estimator that periodically sampled the input stimulus and assumed that environmental variance changed diffusively. The variance estimator has two free parameters. One, $\tau$, represents a prior assumption of the time-scale on which the variance is expected to fluctuate; the other, $\Delta t$, represents the frequency with which the estimator receives new measurements. Measuring more often leads to faster adaptation.

We fed our experimental stimulus into this estimator to determine what an observer would determine the best estimate of the variance to be at each time in the experiment. This variance estimate depended on the stimulus history and the parameters $\tau$ and $\Delta t$.

$$\sigma_{\tau,\Delta t}(t) = \sigma_{est}(x_{stim}(t' < t), \tau, \Delta t) \tag{4}$$

We then calculated the appropriate input rescaling using *Equation 3* and found the parameters

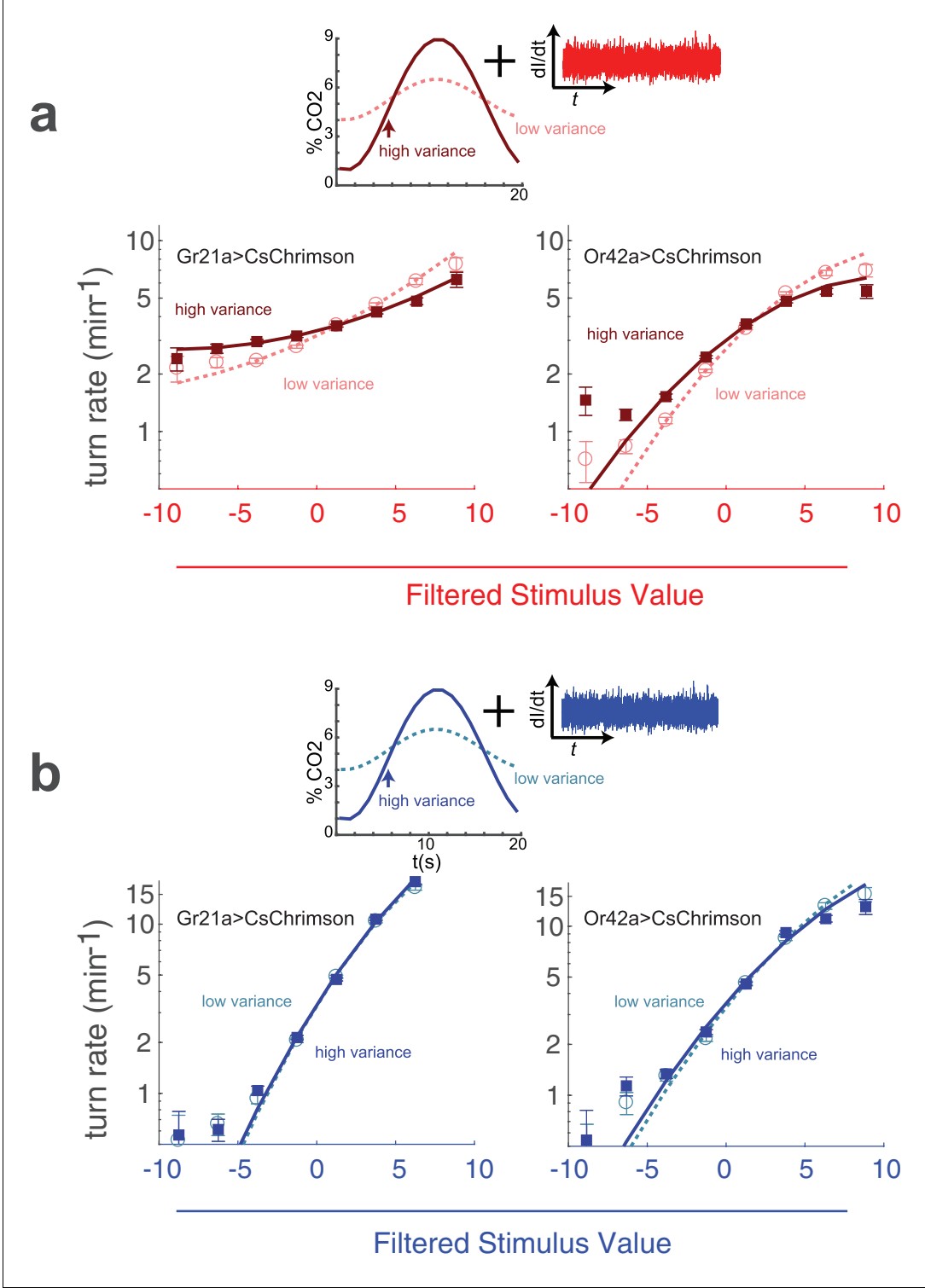

**Figure 9.** Adaptation to variance in a natural odor background. (**a**) Optogenetic activation of odor receptor neurons with a fluctuating background of carbon dioxide. Above: Noise of constant variance was provided to either the $CO_2$ receptor neuron (Gr21a>CsChrimson) or an Ethyl Acetate receptor neuron (Or42a>CsChrimson) in a flow chamber with a time-varying $CO_2$ concentration. The mean $CO_2$ concentration was 5% in both high and low variance conditions, but the amplitude of the fluctuation was larger in the high variance condition. Below: turn rate as a function of filtered stimulus for both high and low variance conditions. Note logarithmic y-axis. (**b**) Visual

*Figure 9 continued on next page*

*Figure 9 continued*

stimulus with a fluctuating background of carbon dioxide. The same as (a) but the stimulus was blue noise of constant variance. *Number of experiments, animals in Table 1*

DOI: https://doi.org/10.7554/eLife.37945.015

of the static rate function that maximized the log-likelihood (Materials and methods) of the observed sequence of turns:

$$\ln(P(\text{data}|\tau,\Delta t)) = \sum_{\text{turn}} \ln(r_{pred}(t;\Delta t,\tau)dt) - \sum_{\text{no turn}} r_{pred}(t;\Delta t,\tau)dt \tag{5}$$

with

$$r_{pred}(t;\Delta t,\tau) = \lambda\left(\alpha(\sigma_{\tau,\Delta t}(t)) \cdot x_{filt}(t)\right) \tag{6}$$

and dt = 1/20 second, the interval at which we sampled the behavioral state.

From these calculations, we found the values of $\tau$ and $\Delta t$ that maximized the likelihood of the data: For visual response these were $\Delta t = 0.85 \pm .07$ s and $\tau = 6 \pm 2.5$ s, and for Or42a activation, these were $\Delta t = .7 \pm .15$ s and $\tau = 11 \pm 5$ s (95% confidence intervals, dotted white regions in *Figure 6a*).

We then compared the predicted rescaling vs. cycle time for the best fit model to the observed rescaling (*Figure 6b*) and found close agreement between the predicted and measured rescalings. When we compared the predictions produced by estimators with the same value of $\tau$ but different values of $\Delta t$, we found substantial disagreement between the predicted and measured rescalings (*Figure 6b*). Thus, the larva's adaptation to environmental variance is consistent with an optimal estimate of that variance and indicates an input sampling rate of ~ 1.2–1.5 Hz.

## Larvae adapt to variance at multiple levels in the navigational pathway

Previously, we studied how larvae combine visual and olfactory information to make navigational decisions. We found that in response to multisensory input, the turn rate was a function of a single linear combination of filtered odor and light signals (*Gepner et al., 2015*). We also found that larvae used quantitatively the same linear combination of light and odor to make decisions controlling turn size and turn direction, suggesting that light and olfactory signals are combined early in the navigational pathway. Here, we asked whether adaptation to environmental variance occurs upstream or downstream of this combination.

To determine the locus of variance adaptation, we presented visual and olfactory white noise inputs simultaneously. The inputs were uncorrelated with themselves or each other; one input remained at constant variance while the other periodically switched between low and high variance. We analyzed the resulting behavior assuming that the turn rate was a nonlinear function of a linear combination of filtered light ($x_L$) and odor ($x_o$), each of which was subject to independent input rescaling

$$r(t,x_o,x_L) = \lambda(\alpha(t)x_o, \beta(t)x_L) \tag{7}$$
$$\lambda(x_o,x_L) = \lambda_0 \exp(ax_o + bx_L) \tag{8}$$

If the larva adapted to variance in each sense individually, we would expect that only the stimulus whose variance was changing would be subject to adaptation, for example for changing odor variance, we would expect $\alpha$ to vary in time and $\beta$ to be constant. On the other hand, if the larva adapted to the variance of, and hence rescaled, the combined odor and light input, we would expect that whether the odor or light were switching variance, both $\alpha$ and $\beta$ would change. We found that only the response to the switching stimulus adapted, while for the fixed-variance stimulus, the turn-rate remained constant (*Figure 7*), indicating that variance adaptation is accomplished on a uni-sensory basis and prior to multi-sensory combination.

Note that we have simplified the rate function to be an exponential of a linear (rather than quadratic) function of $x_o$ and $x_L$. This choice is motivated in the next section, but the same conclusion -

that larvae adapt to variance on a unisensory basis - is reached if a quadratic combination of odor and light is used instead.

We next asked whether variance adaptation was a unique feature of the sensory systems or a more general feature of the navigational circuit. To probe whether variance adaptation can occur in the portion of the navigational circuit that processes combined odor and light information, we presented larvae with visual and olfactory stimuli of fixed variance, but changed the variance of the combined input.

To create a stimulus with constant *uni-sensory* variance but changing *multi-sensory* variance, we presented random white noise visual and olfactory stimuli of constant variance but altered the correlation between them. In the 'high variance' condition, changes in odor and light were negatively correlated (correlation coefficient = −0.8), so unfavorable decreases in odor coordinated with unfavorable increases in light. In the 'low variance' condition, changes in odor and light were positively correlated (correlation coefficient = +0.8), so that favorable increases in odor were coupled with unfavorable increases in light. A circuit element that received only light or odor inputs would be unable to distinguish between the 'high' and 'low' variance conditions, but an element that received a sum of light and odor inputs would observe a much more dynamic environment in the 'high variance' condition.

To analyze these experiments, we created a rotated coordinate system based on $x_o$ and $x_L$, the outputs of the odor and light filters (*Gepner et al., 2015*) (*Figure 8—figure supplement 1*),

$$u = (\cos(\theta)x_o + \sin(\theta)x_L) \tag{9}$$

$$v = (-\sin(\theta)x_o + \cos(\theta)x_L) \tag{10}$$

$\theta$ is a parameter that controls the weighting of odor and light along the two axes of the rotated coordinate system. We constructed the stimulus so that for $\theta = 45°$, $u$ and $v$ were uncorrelated, and had oppositely changing variances with $\sigma_{high}^2 = 9\sigma_{low}^2$. We then analyzed the resulting behavior as before, assuming the turn rate was a nonlinear function of $u$ and $v$, each of which was subject to independent input rescaling

$$r(t,u,v) \equiv \lambda(\alpha_u(t)u, \alpha_v(t)v) \tag{11}$$

$$\lambda(x_o,x_L) = \lambda_0 \exp(au + bv) \tag{12}$$

As in our previous work (*Gepner et al., 2015*), we found a value of $\theta$ (38°) for which $b \approx 0$ for the entire data set. For this value of $\theta$, $b \approx 0$ in both the low and high variance conditions.

Because $v$ had negligible influence on the turn decision, there was no way to determine $\alpha_v$. We found that $\alpha_u$ varied with time, increasing when the variance of $u$ was lower, although the adaptation was less pronounced than for the uni-sensory variance changes (*Figure 7*). This adaptation must be implemented by circuit elements that have access to both light and odor inputs, showing that elements of the navigational circuit well downstream of the sensory periphery also have the ability to adapt to variance.

## Input rescaling likely occurs near the sensory periphery; sense-specific variance adaptation suggests multiplication as a mechanism for multisensory integration

When presented with both light and fictive odor cues, larvae adapt to the variance of each sense individually prior to multi-sensory combination (*Figure 7*). We previously found that larvae *linearly* combine odor and light stimuli to make navigational decisions (*Gepner et al., 2015*). Taken together, these results require a form of variance adaptation that preserves linearity.

Measurement of variance is an inherently nonlinear computation, and we are not aware of any biophysical model of variance adaptation that preserves a linear representation of the stimulus. Adaptation to variance can be achieved by rescaling the output of a rectifying nonlinearity (*Ozuysal and Baccus, 2012*) or by summation of saturating nonlinearities with correlated inputs (*Nemenman, 2012*; *Borst et al., 2005*). A modified Hodgkin-Huxley model of salamander retinal ganglion cells (*Kim and Rieke, 2003*; *Kim and Rieke, 2001*) showed that scaling of the linear filter (equivalent to scaling the input to the nonlinear function) resulted from adaptation of slow Na

+ channels to the mean firing rate of the neuron. While this model linearly rescales the filter, it requires feedback from the nonlinear output of the neuron.

Rectified signals encoding opposite polarities can be combined to produce a linear response (*Werblin, 2010*; *Molnar et al., 2009*), but there is no evidence of such competing pathways in the larva's olfactory system. The adult fly adapts to variance beginning in the olfactory receptor neurons themselves (*Gorur-Shandilya et al., 2017*), making it further unlikely that variance adaptation in the larva is accomplished through rescaling of downstream opponent pathways.

How does the larva adapt to variance through a nonlinear process, prior to apparently combining sensory signals *linearly*? We can resolve this puzzle by changing our model of multisensory integration. In our previous work (*Gepner et al., 2015*), we considered a model in which odor and light turning decisions were mediated by entirely separate pathways, both modeled as LNP processes (*Figure 8a*).

$$\lambda_2(x_o, x_L) = \lambda_o(x_o) + \lambda_L(x_L) \tag{13}$$

This model would be easy to reconcile with modality-specific variance adaptation, but it was far less consistent with our observations of larvae's responses to multisensory input than a competing model (*Figure 8b*) that described the two-input rate function $\lambda_2(x_o, x_L)$ as a one-input function of a linear combination of odor and light

$$\lambda_2(x_o, x_L) = \lambda_1(ax_o + bx_L) \tag{14}$$

While this model makes predictions that are consistent with our multisensory white noise and step experiments, to be consistent with our multisensory variance adaptation experiments, it requires an unknown form of variance adaptation that preserves linearity.

We propose an alternate model (*Figure 8c*), that the two-input rate function might be the *product* of two unisensory rate functions, as recently found in human psychophysical experiments (*Parise and Ernst, 2016*).

$$\lambda_2(x_o, x_L) = \lambda_o(x_o)\lambda_L(x_L) \tag{15}$$

Multiplicative multisensory integration would allow unisensory adaptation to variance through a rectifying nonlinearity while preserving a fundamental feature we observed in multisensory decision making, the ability of favorable changes in one stimulus modality to compensate for unfavorable changes in the other.

In this work and previously (*Gepner et al., 2015*), we modeled the rate function as an exponential of a polynomial function of the filtered input. For a linear polynomial, there is no difference between adding odor and light inputs prior to the exponential nonlinearity and multiplying two exponential functions

$$\lambda(x_o, x_L) = \lambda_0 \exp(ax_o + bx_L) = \lambda_0 \exp(ax_o)\exp(bx_L) \tag{16}$$

In analyzing the multisensory experiments of *Figure 7*, we therefore modeled the rate as an exponential of a linear combination of odor and light, to avoid favoring one model of integration over the other.

For exponentials of a quadratic polynomial, like the 'ratio-of-gaussians' (*Pillow and Simoncelli, 2006*) we used previously, there are quantitative differences between adding the rate function inputs and multiplying rate function outputs, but these can be small, especially if the polynomials are nearly linear over the range of inputs provided in the experiment. We reanalyzed the data of *Gepner et al. (2015)* to determine if multiplicative integration could describe the results as well. While the two models made very similar predictions, we found that for both white noise (*Figure 8—figure supplement 1*) and step experiments (*Figure 8—figure supplement 2*), the multiplicative model produced a statistically significantly better fit to the data than did the early linear combination model, even after accounting for the different number of model parameters. The multiplicative model also better predicted held-out data. Thus, the multiplicative model is at least as explanatory as the early linear combination model for multisensory experiments with constant variance inputs, and it can be reconciled more easily with our finding that larvae adapt to the variance of a sensory input prior to

multisensory combination. Taken together, these results suggest multiplication as a mechanism for multisensory integration.

## Larvae adapt to variance of natural odor backgrounds

In all experiments presented so far, we explored variance adaptation using blue light to present a natural visual stimulus and red light to create fictive olfactory stimuli via optogenetic activation of sensory neurons. Although adaptation to variance of blue light clearly represents a natural response of the visual system, we might wonder to what extent adaptation to variance of fictive stimuli reflects particular properties of the optogenetic channel or peculiarities in fictive rather than natural sensory transduction. We therefore sought to directly measure whether larvae adapt to variance in a natural odor environment.

Although it is difficult to 'flicker' an odor across an extended arena with the speed and precision required for reverse correlation, it is more straightforward to create an odor background of known variance, even if the actual gas concentrations at any particular point in space and time are unknown. We placed larvae in a flow chamber with a constant stream of carrier air and a varying amount of carbon dioxide injected at the inlet. The resulting concentration fluctuations propagated across the arena while being attenuated somewhat by diffusive mixing. Because the flow was laminar, the entire system could be described by linear equations, and the average fluctuation of carbon dioxide at any point in the arena was therefore linearly dependent on the magnitude of the fluctuations at the inlet. We varied the concentration of $CO_2$ at the inlet in a sinusoidal pattern

$$[CO_2] = \mu + A\sin(2\pi t/\tau) \tag{17}$$

In all our experiments, $\mu$, the mean concentration, was $5\%$, and $\tau$, the period of oscillation was 20 s. In the low variance condition, $A$, the amplitude of the oscillation was $1\%$, while in the high variance condition $A = 4\%$. In both conditions, we presented identical red light inputs of constant variance to larvae expressing CsChrimson in $CO_2$ receptor neurons. We then asked whether the larva's response to the fictive stimulus depended on the variance in the background concentration of natural carbon dioxide.

Using our standard reverse-correlation analysis, we extracted the linear kernel (which was the same in both low and high variance conditions) and nonlinear rate function. We found that the rate function was steeper in the low variance context ($\alpha_{high}/\alpha_{low} = 1.56 \pm 0.07$), indicating that a fluctuating background of $CO_2$ reduces the larva's sensitivity to *optogenetic* activation of the $CO_2$ receptor neurons (*Figure 9a*).

We next asked whether changing the background variance of $CO_2$ would influence the response of the larva to activation of the Or42a receptor neuron, which does not respond to carbon dioxide. We found that a higher variance $CO_2$ reduced the response of larvae to activation of the Or42a receptor neurons ($\alpha_{high}/\alpha_{low} = 1.35 \pm 0.05$), but this effect was less pronounced than for activation of the $CO_2$ receptor neurons (*Figure 9a*).

While it is sensible that variation in $CO_2$ levels should affect the sensitivity of the larva to optogenetic perturbation of the $CO_2$ receptor neuron, it is somewhat surprising that $CO_2$ variation altered the response to perturbation of an olfactory neuron that does not respond to $CO_2$. This effect might be due to interactions between the Gr21a and Or42a neurons or pathways, for example lateral presynaptic inhibition (*Olsen and Wilson, 2008*). Or the variation in response to optogenetic activation of the sensory neurons we measured could be due to other effects than variance adaptation -for example, a mathematical result of combining an oscillating signal with white noise prior to a rectifying nonlinearity.

Larvae did not modify their visual sensitivity in response to changes in fictive odor variance (*Figure 7*). We therefore expected that if the effect we observed were due to variance adaptation, it would be absent for white noise visual stimuli combined with natural odor backgrounds. Indeed, when we presented larvae with white noise visual stimuli in a fluctuating $CO_2$ background, we found that the visual response was insensitive to the magnitude of the fluctuations (*Figure 9b*).

## Discussion

In behavioral reverse correlation experiments, we analyzed the output of the entire sensory motor pathway using techniques developed to characterize sensory coding. Despite measuring a behavioral output rather than neural activity in early sensory neurons, we resolved features previously described in sensory systems, including adaptation to variance by input-rescaling, temporal dynamics of adaptation consistent with optimal variance estimators, and stimulus-specific variance adaptation.

### Rates of adaptation suggest that sensory input filters are matched to motor output timescales

In LNP model fits to the reverse-correlation experiments, we found that the convolution kernels for visual and fictive olfactory stimuli have very similar shapes, suggesting that both visual and olfactory stimuli are low passed to the same temporal resolution (*Gepner et al., 2015*). This was somewhat puzzling. In a natural environment and close to surfaces, we would expect that light levels could change much faster than could odor concentrations, whose kinetics would be determined by diffusion and laminar flow. Shouldn't the larva therefore process light information *faster* than odor information?

One resolution to this puzzle is that in behavioral reverse correlation we are not directly measuring the sensory response, but instead the sensory response convolved with the larva's motor output and further limited by the temporal resolution of our video analysis. Therefore, the measured shape of the filter could simply reflect the latency in the motor output pathway or in our analysis software.

But there are reasons to believe that the filter kernels reflect the true temporal dynamics of the sensory systems. The decision to cease a run and end a turn is a navigational response to stimulus changes generated by larva's movement through the environment. For this decision, the larva should seek sensory input that changes at the frequency of its own peristaltic movement, and this time scale should be the same for light and odor.

More generally, there is little value in making a decision faster than it can be implemented by the motor output pathway and a cost (less accuracy) to higher bandwidth measurements. These arguments would suggest that the timescales measured in reverse correlation should be similar to the actual timescales of the sensory neuron filters. Indeed, electrophysiological (*Schulze et al., 2015*; *Gorur-Shandilya et al., 2017*) and optical (*Si et al., 2017*) recordings in olfactory sensory neurons reveal that these neurons filter odor inputs with similar dynamics to those observed in our behavioral reverse correlation experiments.

Variance adaptation provides an independent measure of the bandwidth of the sensory input process. The time it takes larvae to adapt following a switch from high to low variance is long compared to the latencies of the motor output pathway and our analysis software, and, if the larvae's estimates of variance are optimal, determined by the input sampling rate. If light and odor were sampled at different rates, we would expect different rates of adaptation following a switch to low variance. In fact, we see the same rates of adaptation for both light and odor, and these rates are consistent with an optimal estimator sampling the stimulus at a similar frequency to the bandwidth of the convolution kernels. These suggest that sensory input is indeed filtered to match the frequency of the larva's own motion.

If the bandwidth of the filter kernels is set by the speed of the larva's own motion and not the temporal dynamics of the environment *per se*, this would also explain why the temporal structures of the kernels do not adapt to changes in variance (*Figure 1d*, *Figure 3a*).

### Conclusion

In this study, we used behavioral reverse correlation to measure adaptation to environmental variance in a complete sensory-motor transformation. We found that larvae adapted to the variance of visual and fictive olfactory stimuli, that the rate of adaptation was consistent with an optimal estimate of the variance, and that larvae adapted to the variance in each sensory input individually. These results suggest a novel model of multisensory integration in the larva: multiplication of nonlinear representations of sensory input rather than addition of linear representations.

While variance adaptation has been well studied in early sensory neurons, the study of variance adaptation in complete sensory-motor transformations is in its early stages. This work contributes a

study of variance adaptation in a navigational decision-making task as well as behavioral analysis of multi-sensory variance adaptation. Because the larva is transparent and amenable to genetic manipulation, the methods we developed here can be applied to optogenetic manipulation of inter-neurons as well, to understand whether variance adaptation to neural activity is a general feature of neural circuits or specific to early sensory inputs.

Changing the variance of background $CO_2$ levels changes the larva's response to activation of the $CO_2$ receptor neuron but not its response to activation of the photoreceptors, suggesting that variance adaptation to a natural stimulus coupled with optogenetic activation of putative intermediate interneurons might be used to trace the flow of information through neural circuits.

# Materials and methods

## Key resources table

| Reagent type (species) or Resource | Designation | Source or reference | Identifiers | AdditionalInformation |
|---|---|---|---|---|
| Strain (*Drosophila melanogaster*) | Berlin wild type | gift of Justin Blau, NYU | | |
| Genetic reagent (*D. melanogaster*) | w1118;;20XUAS-CsChrimson-mVenus | Bloomington Stock Center | RRID:BDSC_55136 | |
| Genetic reagent (*D. melanogaster*) | w*;;Gr21a-Gal4 | Bloomington Stock Center | RRID:BDSC_23890 | |
| Genetic reagent (*D. melanogaster*) | w*;;Or42a-Gal4 | Bloomington Stock Center | RRID:BDSC_9969 | |
| Genetic reagent (*D. melanogaster*) | w*;;Or42b-Gal4 | Bloomington Stock Center | RRID:BDSC_9972 | |
| Genetic reagent (*D. melanogaster*) | w*;;Or35a-Gal4 | Bloomington Stock Center | RRID:BDSC_9968 | |
| Genetic reagent (*D. melanogaster*) | w*;;Or59a-Gal4 | Bloomington Stock Center | RRID:BDSC_9989 | |
| Software, algorithm | MAGATAnalyzer | (*Gershow et al., 2012*) github.com/samuellab/ MAGATAnalyzer-Matlab-Analysis/ | d9d72b2b43c82af... | |

## Fly strains

The following strains were used: Berlin wild type (gift of Justin Blau), w1118;; 20XUAS-CsChrimson-mVenus (Bloomington Stock 55136, gift of Vivek Jayaraman and Julie Simpson, Janelia Research Campus), w*;;Gr21a-Gal4 (Bloomington stock 23890), w*;;Or42a-Gal4 (Bloomington stock 9969), w*;;Or42b-Gal4 (Bloomington stock 9972), w*;;Or35a-Gal4 (Bloomington stock 9968), w*;;Or59a-Gal4 (Bloomington stock 9989)

## Crosses

About 50 virgin female UAS-CsChrimson flies were crossed with about 25 males of the selected Gal4 line. F1 progeny of both sexes were used for experiments.

## Larva collection

Flies were placed in 60 mm embryo-collection cages (59–100 , Genessee Scientific) and allowed to lay eggs for 3 hr at 25C on enriched food media ('Nutri-Fly German Food,' Genesee Scientific). For all experiments except the Berlin response to blue light (top rows of *Figure 1d,e*; *Figure 2a,b*; top row of *Figure 3*; top row of *Figure 5*; left column of *Figure 6*), the food was supplemented with 0.1 mM all-trans-retinal (ATR, Sigma Aldrich R2500), and cages were kept in the dark during egg laying. When eggs were not being collected for experiments, flies were kept on plain food at 20C.

Petri dishes containing eggs and larvae were kept at 25C (ATR +plates were wrapped in foil) for 48–60 hr. Second instar larvae were separated from the food using 30% sucrose solution and washed in deionized water. Larval stage was verified by size and spiracle morphology. Preparations for experiments were carried out in a dark room, under dim red (for phototaxis experiments) or blue

(for CsChrimson experiments) illumination. Prior to beginning experiments, larvae were dark adapted on a clean 2.5% agar surface for a minimum of 10 min.

## Behavioral experiments

Approximately 40–50 larvae were placed on a darkened agar surface for behavioral experiments (as in *Gepner et al., 2015*). The agar surface resided in a 23 cm square dish (Corning BioAssay Dish #431111, Fisher Scientific, Pittsburgh, PA), containing 2.5% (wt/vol) bacteriological grade agar (Apex, cat #20–274, Genesee Scientific) and 0.75% (wt/vol) activated charcoal (DARCO G-60, Fisher Scientific). The charcoal darkened the agar and improved contrast in our dark-field imaging setup. The plate was placed in a darkened enclosure and larvae were observed under strobed 850 nm infra-red illumination (ODL-300–850, Smart Vision Lights, Muskegon, MI) using a 4 MP global shutter CMOS camera (Basler acA2040-90umNIR, Graftek Imaging) operating at 20 fps and a 35 mm focal length lens (Fujinon CF35HA-1, B&H Photo, New York, NY), and equipped with an IR-pass filter (Hoya R-72, Edmund Optics). A microcontroller (Teensy ++2.0, PJRC, Sherwood, OR) coordinated the infrared strobe and control of the stimulus light source, so stimulus presentation and images could be aligned to within the width of the strobe window (2–5 ms). Videos were recorded using custom software written in LABVIEW and analyzed using the MAGAT analyzer software package (*Gershow et al., 2012*; https://github.com/samuellab/MAGATAnalyzer-Matlab-Analysis). Further analysis was carried out using custom MATLAB scripts. *Table 1* gives the number of experiments, animals, turns, and head sweeps analyzed for each experimental condition.

To determine the number of experiments to perform, we did a back-of-the-envelope calculation, assuming that we would find the time-varying rate function by histogram division, and set 30 experiments as the target for the visual switching experiment of *Figure 3*. When we actually analyzed the data using more sophisticated methods, we found that fewer experiments would yield adequate signal to noise. We then aimed for at least 10 experiments for each condition, although we did more or (in one instance) fewer depending on the fecundity of the flies. For *Figure 9*, only a single constant rate function is extracted using all the available data, so fewer experiments could be performed in each condition.

All replicates are biological replicates, except for the jackknife predictions of held-out data in *Figure 2* and *Figure 8—figure supplement 1* and bootstrapping to produce the shaded error regions on the turn-triggered averages.

## Stimuli

### Light intensity

Stimuli were provided by changing the intensity of blue (central wavelength $\lambda = 447.5$ nm) and red ($\lambda = 655$ nm) lights on a custom-built LED board (*Gepner et al., 2015*) for visual and fictive olfactory signals respectively. The red light intensity varied from 0 to $900 \frac{\mu W}{cm^2}$. For unisensory visual experiments, the blue light intensity varied from 0 to $50 \frac{\mu W}{cm^2}$. For multi-sensory experiments (*Figure 7*), the blue light intensity varied from 0 to $3 \frac{\mu W}{cm^2}$.

### Light sequences

Light levels were specified by values between 0 (off) and 255 (maximum intensity). These values changed according to a Brownian random walk (*Gepner et al., 2015*), whose derivatives on all time scales are independent identically distributed Gaussian variables.

The light level was set by pulse width modulation and updated every $1/120$ s. $I_j$ represents the intensity at time $t_j = j/120$. For all experiments except those of *Figure 7* (bottom row), sequences of light levels were generated according to these rules:

$$I_0 = 127 \tag{18}$$
$$I_{j+1} = I_j + N(0, \sigma) \tag{19}$$
$$I_j = -I_j \text{ if } I_j < 0 \tag{20}$$
$$I_j = 510 - I_j \text{ if } I_j > 255 \tag{21}$$

where $N(0, \sigma)$ was a Gaussian random variable with mean 0 and variance $\sigma^2$. The sequence was generated in floating point (i.e. non-integer values were allowed) and converted to integers for output.

We analyzed behavioral responses to light intensity derivatives, so the input variance was dictated by $\sigma^2$. For low-variance conditions, $\sigma^2 = 1$ and for the high variance condition $\sigma^2 = 9$. In the experiments displayed in the top two rows of **Figure 7**, the non-switching stimulus was generated with $\sigma^2 = 4$, although the same conclusions were reached when the non-switching stimulus was generated with $\sigma^2 = 1$ or 9. In **Figure 9**, $\sigma^2 = 9$ for all stimuli.

Sequences were not reused within an experimental group but might be reused between groups.

For multi-modal experiments in **Figure 7** (top and middle), independent sequences were used for each stimulus.

For the experiments with uncorrelated blue light intensities (**Figure 4**), a new light level was selected every 0.25 s. In the low variance condition, these were drawn from a normal distribution with mean 128 and standard deviation 17. In the high variance condition, the distribution had mean 128 and standard deviation 51. Out of range values (below 0 or above 255) were adjusted to 0 or 255 appropriately.

## Correlated multisensory sequences

For multi-modal experiments in **Figure 7** (bottom), we generated two correlated Brownian sequences $I^o$ and $I^l$:

$$I^o_{j+1} = I^o_j + s^o \tag{22}$$

$$I^l_{j+1} = I^l_j + s^l \tag{23}$$

where $s^o$ and $s^l$ were normally distributed with mean 0 and individual variances $\sigma^2$.

The correlation coefficient of the odor and light inputs, $c$, is given by

$$c = \frac{\langle s^o s^l \rangle}{\sqrt{\langle (s^o)^2 \rangle \langle (s^l)^2 \rangle}} \tag{24}$$

$$= \frac{\langle s^o s^l \rangle}{\sigma^2} \tag{25}$$

In previous work, we found that larvae respond to a single combination of filtered odor and light inputs

$$u = \cos(\theta)x_o + \sin(\theta)x_l \tag{26}$$

We did not know *a priori* the exact value of $\theta$ we would find in a variance switching experiment, so we created a stimulus that would have a comparable switch in variance $\sigma^2_{high} = 9\sigma^2_{low}$ for $\theta = 45°$.

The variance of $u = \frac{1}{\sqrt{2}}(s^o + s^l)$ is

$$\sigma^2_u = \langle \frac{1}{2}(s^o + s^l)^2 \rangle = \sigma^2(1+c) \tag{27}$$

so to change the variance between $\sigma^2_u = 9$ and $\sigma^2_u = 1$, we periodically changed the light-odor correlation coefficient between a positive and negative value: $c = \pm 0.8$, and set $\sigma^2 = 5$.

To generate this signal, we chose

$$s^o = \sqrt{1 - |c|} * x_1 + \sqrt{|c|} * x_3 \tag{28}$$

$$s^l = \sqrt{1 - |c|} * x_2 \pm \sqrt{|c|} * x_3 \tag{29}$$

where $|c| = 0.8$ and $x_1$, $x_2$, and $x_3$ are three different and uncorrelated Gaussian random variables with mean 0 and variance $\sigma^2 = 5$. Keeping in mind that the odor kernel has opposite sign to the light kernel, the high variance condition is when $c = -0.8 : s^l = \sqrt{1 - |c|} * x_2 - \sqrt{|c|} * x_3$ and the low variance condition is when $c = +0.8$

## Natural carbon dioxide background

Reverse correlation experiments of *Figure 9* were conducted in a laminar flow chamber with a glass lid through which the behavioral arena could be observed. Mass Flow Controllers (AAlborg) were controlled via Labview to generate a sinusoidally varying $CO_2$ concentration via mixing of pure $CO_2$ and filtered compressed air. Air flow was fixed at 4 L/min and $CO_2$ flow varied sinusoidally, with a period of 20 s, either between 0.12 and 0.22 L/min or between 0 and 0.35 L/min, respectively for low and high variance experiments.

## Data extraction

Videos of behaving larvae were recorded using LabView software into a compressed image format (mmf) that discards the stationary background (*Gershow et al., 2012*; *Kane et al., 2013*). These videos were processed using computer vision software (written in C++ using the openCV library) to find the position and posture (head, tail, midpoint, and midline) of each larva and to assemble these into tracks, each following the movement of a single larva through time. These tracks were analyzed by Matlab software to identify behaviors, especially runs, turns, and head sweeps.

The sequence of light intensities presented to the larvae was stored with the video recordings and used for reverse-correlation analysis.

## Data analysis

For all experiments, we discarded the first 60 s of data to let the larvae's response to a novel environment dissipate. We use $\lambda(x)$ to indicate a static rate function that does not contain a variance adaptation term and $r(x), r(x,t)$ to indicate full rate functions that include variance adaptation.

### Kernels and rate functions

We calculated kernels and rate functions separately for low and high variance by pooling together all the data from low variance and high variance contexts. In *Figure 1* and *Figure 2*, we discarded the first 15 s from each cycle. In *Figures 3* and *7*, we discarded the first 10 s from each cycle.

Filters (*Figures 1d*, *3a* and *4a*): Filters, or kernels, were calculated as the 'turn-triggered' average signal for each set of experiments (*Gepner et al., 2015*). That is, we extracted the sequences of light-intensity derivatives, in bins of 0.1 s, surrounding every turn, and averaged them together. We scaled the low and high variance average signals to have the same maximum value.

For experiments with random intensities (*Figure 4*), we subtracted the mean of the stimulus from the turn-triggered average prior to scaling. To remove artifacts associated with the 4 Hz update rate, we low passed the turn-triggered averages using a Gaussian filter with $\sigma = 0.25s$.

To calculate the shaded error regions, we adopted a bootstrapping approach (*Zhou et al., 2018*). For each set of experiments (e.g. the 17 blue light experiments of *Figure 1*):

1. We generated a resampled data set by randomly selecting experiments and larvae with replacement
   1. we selected, with replacement, a random subset of equal length. For example, if there were four experiments, a valid subset might be (1,1,3,2)
   2. For each experiment in this subset, we selected, with replacement, a random subset of individual larvae.
2. We calculated the turn-triggered average of this resampled data set to create a single bootstrapped average.
3. We repeated the above steps 100 times
4. The shaded error region is the standard deviation at each time bin of the 100 bootstrapped averages.

Compared to simply calculating the standard error of the mean for each time bin, this approach respects the possibility of correlated sources of noise in the experiments.

Direct estimation of turn rates (*Figures 1e*, *3b*, *7* and *9*): The turn rates (in $min^{-1}$), were calculated from the data as:

$$r(x_f) = 60 \cdot \frac{N_{turn}(x_f)}{N_{run}(x_f)} \cdot \frac{1}{\Delta t} \tag{30}$$

$$\sigma_r(x_f) = 60 \cdot \frac{\sqrt{N_{turn}(x_f)}}{N_{run}(x_f)} \cdot \frac{1}{\Delta t} \tag{31}$$

where $\Delta t = \frac{1}{20} s$ was the sampling period. $N_{turn}(x_f)$ is the number of turns observed with the filtered signal $x_f$ within one of the $n = 8$ bins containing $x_f$. $N_{run}(x_f)$ is the total number of data points where the filtered signal was in the histogram bin and larvae were in runs and thus capable of initiating turns.

## Maximum likelihood estimation of static turn rates

For *Figures 1e*, *3b* and *9*, we separately fit high and low variance rate functions to exponentials of quadratics:

$$r(x) = \begin{cases} \lambda_0^{(h)} \exp(b^{(h)}x + c^{(h)}x^2) & \text{high variance} \\ \lambda_0^{(l)} \exp(b^{(l)}x + c^{(l)}x^2) & \text{low variance.} \end{cases} \tag{32}$$

with $\lambda_0^{(h)}, \lambda_0^{(l)}$, $b^{(h)}, b^{(l)}$, and $c^{(h)}, c^{(l)}$ as independent parameters.

The probability of observing at least one turn in an interval $\Delta t$ given an underlying turn rate $r$ is $1 - e^{-r\Delta t}$. In the limit of short $\Delta t$ this reduces to $r\Delta t$. The probability of not observing a turn in the same interval is $e^{-r\Delta t}$.

For a model of the turn rate, the log-likelihood of the data given that model is therefore

$$\log(P(data|model)) = \sum_{turn} \log(r(x)\Delta t) - \sum_{no\ turn} r(x)\Delta t \tag{33}$$

where $x$ is the filtered signal, $r(x)$ is the turn rate predicted by the model, and $\Delta t = \frac{1}{20} s$ is the sampling rate in our experiments. $\sum_{no\ turn}$ is a sum over all points when larvae were in runs and thus capable of initiating turns.

We used the MATLAB function fminunc to find the parameters that maximize this log-likelihood. *Figure 7b,c* were fit in the same fashion but for exponentials of linear functions ($c \equiv 0$).

## Comparison of rescaling models (*Figure 2*)

For the experiments in *Figure 1d–e*, we model the rate function as: $\lambda(x) = \lambda_0 \exp(bx + cx^2)$, and ask if adaptation is better characterized by:

- an input rescaling:

$$r(t,x) = \lambda(\alpha(t) \cdot x)$$

- an output rescaling:

$$r(t,x) = \alpha(t) \cdot \lambda(x)$$

- or a 're-centered' output rescaling (where the basal rate is adjusted after rescaling):

$$r(t,x) = \alpha(t) \cdot (\lambda(x) - \lambda_0) + \lambda_0$$

For each model, we fit low and high-variance rate functions simultaneously (excluding the first 15 s of each cycle) by minimizing the negative-log-likelihood. We set $\alpha_{high} = 1$ and thus have four fit parameters for each model ($\lambda_0$, $b$, $c$, and $\alpha_{low}$). We plot the resulting low and high-variance rate functions for each model (*Figure 2a,d*), and also fit the data to a null model with no adaptation (only three fit parameters: $\lambda_0$, $b$, $c$), for which the low and high-variance rate functions are identical. We find that the input rescaling is the best model of the larvae's turn-rate adaptation.

We then fit a subset (14/17 experiments) of the data to each model and find how well each one predicts the remainder of the data (3/17 experiments). By permuting the fitted and tested portions of the data, we find jackknife estimates of the log-likelihood of the test-data given the fit rate function (mean and standard error shown in *Figure 2b,e*). We then compare the input and recentered output models directly, showing the histogram of log-likelihoods for all different permutations of fitted and tested portions (*Figure 2c,f*).

In **Figure 2b,c** 20/680 and in **Figure 2d,f** 97/680 jack-knives resulted in the recentered output model producing a negative rate for some of the test data. These were excluded from analysis.

Assuming the entirety of a test data set is generated by the same process, we would expect that the log-likelihood of that data set given any model would be proportional to the length of the data set. To normalize out the effects of fluctuations in the number of animal-minutes in the held-out data sets, we therefore normalized the log-likelihood

$$\bar{LL}_j = LL_j * \langle T \rangle / T_j \tag{34}$$

where $LL_j$ is the likelihood of the test data in the $j^{th}$ jack-knife, $T_j$ is the total observation time in the $j^{th}$ test data set, and $\langle T \rangle = \frac{1}{N}\sum_{j=1}^{N} T_j$.

## Bayes information criterion

The Bayes Information Criterion (BIC) is defined as

$$\log(n_{data}) * n_{params} - 2 * \log(P(\text{data}|\text{model})) \tag{35}$$

BIC is closely related to the Aikake Information Criterion (AIC) but more strongly favors models with fewer parameters. $\Delta BIC < -10$ strongly favors the more negative model (roughly equivalent to $p<0.01$).

We chose $n_{data}$, the sample size to be the total number of larva-seconds. If we had chosen $n_{data}$ to be the total number of larvae, $\Delta BIC$ would shift to favor models with more parameters by about seven per extra parameter. If we had chosen $n_{data}$ to be the total number images of individual larva analyzed, then $\Delta BIC$ would shift to favor models with fewer parameters by 3 ($\log 20$) per extra parameter. Neither of these changes would have affected which model fits were preferred or the significance of the differences between models.

## Calculation of $\alpha(t)$ (**Figures 3c**, **5a** and **7a**)

For uni-sensory experiments (**Figures 3c** and **5a**), we characterized adaptation of the turn rate as an input rescaling:

$$r(t,x) = \lambda(\alpha(t) \cdot x(t)) \tag{36}$$

where $x$ is the (convolved) stimulus value, and $\lambda$ is a fixed nonlinear function:

$$\lambda(x) = \lambda_0 \exp(bx + cx^2) \tag{37}$$

Our goal is to find the values of $\lambda_0$, $b$, $c$, and $\alpha(t)$ that maximize the probability of the data and a prior probability that constrains the smoothness of $\alpha$. We grouped all experiments together and found a single $\alpha(t)$ for each time point between 60 s and 20 min; we did not use any prior knowledge of variance switching times in the extraction of best-fit parameters. We use an iterative process to find the best fit model parameters.

First, we assume that $\alpha \equiv 1$, ignoring adaptation to variance, and calculate the best fit parameters of a static rate function, $\lambda(x) = \lambda_0 \exp(bx + cx^2)$, using maximum likelihood estimation described above.

Next, we calculate the time-varying scaling parameter $\alpha(t)$. At each time point $t_i$, we use Bayes' rule to calculate the probability of a given value of $\alpha$

$$P(\alpha_i | data_{j \leq i}) = \frac{1}{\Omega} P(data_i | data_{j<i}, \alpha_i) \cdot P(\alpha_i | data_{j<i}) \tag{38}$$

In the LNP formulation, we assume that turns are generated by a memoryless Poisson process, so we simplify $P(data_i | data_{j<i}, \alpha_i) = P(data_i | \alpha_i)$. We assume that the prior probability $\alpha_i$ of alpha depends only on $\alpha_{i-1}$, so $P(\alpha_i | \alpha_{j<i}) = P(\alpha_i | \alpha_{i-1})$. We can then write

$$P(\alpha_i | data_{j \leq i}) = \frac{1}{\Omega} P(data_i | \alpha_i) \cdot \int d\alpha_{i-1} P(\alpha_i | \alpha_{i-1}) P(\alpha_{i-1} | data_{j<i}) \tag{39}$$

where $P(\alpha_{i-1} | data_{j<i})$ is the distribution of $\alpha$ values at the previous time step, and $P(data_i | \alpha_i)$ is the

likelihood of seeing the observed turn times, $data_i$, given that they are generated by a Poisson process with a mean rate $r_i$:

$$P(data_i|\alpha_i) = \prod_{turn} r_i(x)\Delta t \prod_{no\ turn} \exp(-r_i(x)\Delta t) \tag{40}$$

where $r_i(x) = \lambda(\alpha_i x)$, using the fixed static rate function calculated previously. For $P(\alpha_i|\alpha_{i-1})$, we chose a Gaussian prior

$$P(\alpha_i|\alpha_{i-1}) = \frac{1}{\sqrt{4\pi\Delta t/\tau}}\exp(\frac{-(\alpha_i - \alpha_{i-1})^2}{4\Delta t/\tau}) \tag{41}$$

$\Delta t$ is the time-step of the fitting routine and $\tau$ controls the smoothness of our estimate of $\alpha$. $\Delta t = 0.1s$ in **Figure 3c** and $\Delta t = 1s$ in **Figure 5a**, and with $\tau = 5s$ for both.

Using this formulation, we calculate the most likely value of $\alpha_i$ at each time point $t_i$ ($\alpha(t)$) and the uncertainty in this estimate ($\sigma(t)$). We use the new values of $\alpha(t)$ to find new best fit parameters of the static rate function and iterate the process until the log-likelihood converges.

Finally, to describe how $\alpha$ varies in response to changes in variance, we calculated an appropriately weighted average over cycle time. Call the time of the start of each cycle (e.g. switch to low variance) $t_k^{switch}$, then

$$\alpha(\tau) = \sum_k w_k \alpha(t_k^{switch} + \tau) \tag{42}$$

$$w_k = \frac{1}{\sigma(t_k^{switch} + \tau)}/\sum_k \frac{1}{\sigma(t_k^{switch} + \tau)} \tag{43}$$

To verify that the asymmetry in the timescales behavior is not an artifact of our analysis, we generate artificial turn-decisions from a rate-function that switches instantaneously when the variance switches. We then estimate the corresponding scaling factors for these new data and find that our estimator tracks the adaptation equally quickly for upward and downward switches (black curves in **Figure 3c**).

For multi-sensory experiments (**Figure 7a**), the rate is modeled as

$$r(x_o, x_l) = \lambda(\alpha(t) \cdot x_o(t) + \beta(t) \cdot x_l(t))$$

Two dimensional, $(\alpha, \beta)$ distributions are calculated at each time step:

$$P(\alpha_i, \beta_i|data_{j\leq i}) = \frac{1}{\Omega}P(data_i|\alpha_i, \beta_i) \cdot P(\alpha_i, \beta_i|data_{j<i}) \tag{44}$$

with

$$P(\alpha_i, \beta_i|data_{j<i}) = \int d\alpha_{i-1}d\beta_{i-1}P(\alpha_i, \beta_i|\alpha_{i-1}, \beta_{i-1})P(\alpha_{i-1}, \beta_{i-1}|data_{j<i}) \tag{45}$$

The prior is now a two-dimensional Gaussian:

$$P(\alpha_i, \beta_i|\alpha_{i-1}, \beta_{i-1}) = \frac{1}{2\pi\sqrt{|\Sigma|}}\exp\left(-\frac{1}{2}(\overrightarrow{\alpha} - \overrightarrow{\mu})^T\Sigma^{-1}(\overrightarrow{\alpha} - \overrightarrow{\mu})\right) \tag{46}$$

where

$$\overrightarrow{\alpha} = \begin{bmatrix} \alpha \\ \beta \end{bmatrix} \tag{47}$$

$$\overrightarrow{\mu} = \begin{bmatrix} \mu_\alpha \\ \mu_\beta \end{bmatrix} = \begin{bmatrix} \alpha_{i-1} \\ \beta_{i-1} \end{bmatrix} \tag{48}$$

$$\Sigma^{-1} = \frac{1}{2\Delta t}\begin{bmatrix} \tau_\alpha & \tau_{\alpha\beta} \\ \tau_{\alpha\beta} & \tau_\beta \end{bmatrix} \tag{49}$$

and $|\Sigma|$ is the determinant of $\Sigma$.

$\Delta t = 0.1s$ $\tau_\alpha = \tau_\beta = 5s$, and $\tau_{\alpha\beta} = 0$ so that no correlation between changes in $\alpha$ and $\beta$ was introduced in our prior expectation.

To find $\alpha(t)$ and $\beta(t)$, we marginalize the $(\alpha_i, \beta_i)$ distribution at each time step and calculated the most likely value and uncertainty of $\alpha$ and $\beta$ separately.

For the correlated stimuli (*Figure 7a*, bottom row), we first looked for an angle $\theta$ such that the rate function could be characterized by a linear combination of filtered odor and light inputs, $u = (\cos(\theta)x_o + \sin(\theta)x_l)$ (*Gepner et al., 2015*). We thus looked for a model of the form:

$$r(u) = \lambda(\alpha_u u) \tag{50}$$
$$\lambda(u) = \lambda_0 \exp(au) \tag{51}$$

where $\alpha_u$ described the rescaling from high to low variance. We fit low and high-variance rate functions simultaneously (excluding the first 10 s of each cycle) by minimizing the negative-log-likelihood. We set $\alpha_{u,high} = 1$ and thus have four fit parameters ($\lambda_0$, $a$, $\theta$, and $\alpha_{u,low}$). In this way we found $\theta \approx 38°$.

Using this value of $\theta$, we then fit the data to the two-coordinates model:

$$r(u,v) = \lambda(u,v) \tag{52}$$
$$\lambda(u,v) = \lambda_0 \exp(au + bv) \tag{53}$$

with $u$ defined above, $v$ the coordinate orthogonal to $u : v = (-\sin(\theta)x_o + \cos(\theta)x_l)$. There, we found $b \approx 0$.

We could then use the one-coordinate model (*Equation 51*) and the iterative procedure described earlier in this section (*Equation 40*) to calculate $\alpha_u(t)$ using $\tau_u = 5s$ for the correlation time of the prior distribution (*Figure 7a*, bottom row).

## Bayesian-optimal estimates of the stimulus variance (*Figure 6*)

We calculate Bayesian-optimal estimates of the stimulus variance (*DeWeese and Zador, 1998*):

$$\begin{aligned} P(\sigma_i|s_{j\leq i}) &= \frac{1}{\Omega}P(s_i|\sigma_i) \cdot P(\sigma_i|s_{j<i}) \\ &= \frac{1}{\Omega}P(s_i|\sigma_i) \cdot \int d\sigma_{i-1} P(\sigma_i|\sigma_{i-1})P(\sigma_{i-1}|s_{j<i}) \end{aligned} \tag{54}$$

$s_i = I(t_i) - I(t_{i-1})$ is the total change in light level over the sampling interval and forms the input to the estimator. $\sigma_i$ is the estimate of the standard deviation of the light level changes and is the output of the model. We have replaced $P(s_i|s_{j<i})$ with $P(s_i)$ because stimulus samples are uncorrelated.

The sampling time, $\Delta t$, determines the rate at which the estimator picks out samples from the stimulus ensemble to make estimates of the variance. The estimator requires a prior model of how environmental variance changes with time. We chose a diffusive prior, parameterized by a correlation time $\tau$:

$$P(\sigma_i|\sigma_{i-1}) = \frac{1}{\sqrt{4\pi\Delta t/\tau}}\exp(\frac{-(\sigma_i - \sigma_{i-1})^2}{4\Delta t/\tau}) \tag{55}$$

Decreasing $\tau$ increases the speed at which the estimator responds to changes in variance at the cost of decreasing stability during periods of constant variance. In the absence of measurement noise, decreasing $\Delta t$ strictly improves the performance of the estimator, increasing response speed and stability.

To determine the values of $\Delta t$ and $\tau$ that were most consistent with the larva's behavior, we estimated the stimulus variance using a series of estimators with different choices of $\Delta t$ and $\tau$. For each such estimate of the signal variance:

$$\sigma_{\tau,\Delta t}(t) = \sigma_{est}(x_{stim}(t'<t), \tau, \Delta t) \tag{56}$$

we calculated a mapping from variance to rescaling parameter using the relation:

$$\alpha(\sigma) = \frac{\alpha_0}{\sqrt{\sigma^2 + \sigma_0^2}} \tag{57}$$

with $\sigma_0$ taken from the switching experiments of *Figure 3* and $\alpha_0$ chosen to enforce $\langle \alpha(t) \rangle = 1$.

For each set of $\Delta t$, $\tau$, we now had a predicted rescaling for each time point in the experiments: $\alpha_{opt}(t)$. To compare these predictions to the data, we found the log-likelihood of the observed sequence of data given the rate $r(t, x) = \lambda(\alpha_{opt}(t) \cdot x(t))$, with the parameters of the static rate function $\lambda(x) = \lambda_0 \exp(bx + cx^2)$ found separately for each set of $\Delta t$, $\tau$ by maximum-likelihood-estimation.

## New methods developed

Behavioral reverse correlation using visual and optogenetic stimulation follows previous work (*Gepner et al., 2015*; *Hernandez-Nunez et al., 2015*). Behavioral analysis software was developed in *Gershow et al. (2012)*. New to this work are: presentation of stimuli with changing variance; analysis of adaptation to variance, including estimation of the time-varying rescaling parameter, estimation of the rescaling parameter vs. variance, and comparison of the adaptation with predictions of a Bayes estimator; combination of optogenetic and visual manipulation with fluctuating $CO_2$ background; multi-sensory variance adaptation including the use of correlated stimuli of constant variance to produce a multi-sensory signal with changing variance.

## Acknowledgements

We thank Ilya Nemenman for discussions at the Aspen Center for Physics and KITP summer programs on behavior. This work was supported by NIH grant 1DP2EB022359, NSF grant 1455015, and a Sloan Research Fellowship.

## Additional information

### Funding

| Funder | Grant reference number | Author |
| --- | --- | --- |
| National Institutes of Health | 1DP2EB022359 | Jason Wolk<br>Marc Gershow |
| National Science Foundation | 1455015 | Ruben Gepner<br>Jason Wolk<br>Marc Gershow |
| Alfred P. Sloan Foundation | | Marc Gershow |
| National Institutes of Health | R90DA043849 | Sophie Dvali |

The funders had no role in study design, data collection and interpretation, or the decision to submit the work for publication.

### Author contributions

Ruben Gepner, Conceptualization, Data curation, Software, Formal analysis, Investigation, Methodology, Writing—original draft, Writing—review and editing; Jason Wolk, Data curation, Formal analysis, Investigation, Writing—review and editing; Digvijay Shivaji Wadekar, Sophie Dvali, Investigation, Writing—review and editing; Marc Gershow, Conceptualization, Software, Formal analysis, Supervision, Funding acquisition, Methodology, Writing—original draft, Project administration, Writing—review and editing

### Author ORCIDs

Digvijay Shivaji Wadekar http://orcid.org/0000-0002-2544-7533
Marc Gershow http://orcid.org/0000-0001-7528-6101

### Decision letter and Author response

Decision letter https://doi.org/10.7554/eLife.37945.019

Author response https://doi.org/10.7554/eLife.37945.020

## Additional files

### Supplementary files
• Transparent reporting form
DOI: https://doi.org/10.7554/eLife.37945.017

### Data availability
All data generated or analysed during this study are included in the manuscript and supporting files.

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
