## [Decision Letter]

Thank you for submitting your article "Variance Adaptation in Navigational Decision Making" for consideration by *eLife*. Your article has been reviewed by 2 peer reviewers, and the evaluation has been overseen by a Reviewing Editor and Eve Marder as the Senior Editor. The reviewers have opted to remain anonymous.

The reviewers have discussed the reviews with one another and the Reviewing Editor has drafted this decision to help you prepare a revised submission.

Summary:

Gepner et al. present data showing that larval navigational behavior adapts to the variance of various stimuli. They use optical stimuli to probe visual pathways naturally and olfactory pathways via channelrhodopsin; both pathways show contrast adaptation, but not cross-adaptation between the two pathways. (A very elegant experiment with correlated vs. uncorrelated noise between the two channels showed some mild interactions.) They measure the timescale of the adaptation and show that the different kinetics for adapting to contrast increments vs. decrements are consistent with an optimal detection theory. The authors also used a more natural stimulus paradigm to show that this sort of adaptation is not purely a function of their artificial stimuli.

Throughout, the authors fit their data to a variety of models. They showed that their data were best explained by input rescaling, that their data were consistent with a model in which a Bayes' optimal estimate of variance occurs on timescales of ~1s, and that signals are combined with a potentially multiplicative interaction.

The experiments are well done and the analysis is convincing.

Essential revisions:

Two major concerns arose in consultation:

1) The protocol for contrast adaptation looks different from the classical ones in the field, and reviewer #1, point 1 requests clarification and a new experiment that should be doable in a short time.

2) We would like to see a quantification of how filter shape depends on stimulus variance, and reviewer #2, point 1 provides details of the analyses needed.

*Reviewer #1:*

1) If I understand it correctly, the change in stimulus variance is related to a change in correlation time of the light intensity, rather than in its variance about a mean, in order to get the derivatives to scale as intended. Does this cause any problems? What if you just do it the naïve way, by scaling the entire light intensity trace contrast, so that the derivatives and deviations both scale up? Is the answer the same? I think this is potentially important because contrast adaptation is crucially dependent on the timescale on which one computes contrast, and by changing the timescale of the stimulus correlations, one might change regimes. (As an extreme case, if fluctuations were made incredibly slow, one would presumably begin to measure adaptation to the mean, rather than to the contrast.)

2) The authors frame the adaptation as rescaling the nonlinearity, but what if adaptation rescales the linear filter amplitude. These are mathematically equivalent, and if you do the rescaling of the linear filter amplitude, then the Figure 8B model seems like it could be reconciled with the independent adaptation of the two channels. Is this a problem for excluding Figure 8B for this reason?

3) The multiplicative model is different from additive in the case of polynomial terms of order >1 in the exponential. However, if you expand everything in Taylor series, is the multiplication just allowing a few more higher order interactions between the terms? The relative coefficients of those higher order terms is restricted by this multiplication step. But what happens if you fit models that just allow up-to-cubic interactions between the x_O_ and x_L_ filtered terms before applying the exponential? (This would be a nonlinear interaction before applying the second nonlinearity.) Can you do better than multiplication? If so, would that imply that there's some kind of nonlinear summation of the terms with potentially small nonlinearities that is in fact the best fit? Or is multiplicative really a better model because it requires fewer fitting terms? A BIC evaluation seems like it could resolve this.

*Reviewer #2:*

1) A major point of confusion for me was that the authors claim that the linear filters are conserved across different values of stimulus variance. Unless I missed it, there is no quantification of this (except for the claim of 'having established that the kernel shape did not depend on the input variance', subsection “Larvae Adapt Their Turn-Rate to the Variance in Visual and Olfactory Sensory In-puts”), and the figures shown (Figure 1D, Figure 3A) suggest that there actually is a consistent effect of increasing variance on filter shape – namely the relative height of the left shoulder of the filter. While this is not a qualitative change in filter shape, it does change to what extent larvae take into account information about the stimulus further away from the turn. Hence, I would like to see a quantification of how filter shape depends on stimulus variance (e.g., by plotting f_hi vs f_low – if filter shape is preserved across variances, all points should be close to the diagonal – it looks as if filter values close to the peak will lie on the diagonal, but filter values around -6 to -4 seconds prior to the turn should lie off diagonal). Using the actual filter shape for each variance, rather than the filter derived from pooled data, might affect (in both directions) the reported effects of variance on the shape of the nonlinearity.

2) There were also some issues with the writing. For example, the abstract is lacking a description of the niche this work aims to fill, and a statement on the general relevance of the findings – as another example, the authors don't example why it is interesting that multisensory integration involves a multiplicative step and what the implication is for behavior. I also often had to go back to their previous paper (Gepner et al., 2015) and dig into the details of their model or look up details on the coordinate transform (see comment below). The rationale for each plot is not made sufficiently clear to the reader. It would help if the authors show more raw data to provide an intuition for the derived plots – e.g., what happens to animal turning following the switch in variance from low to high or high to low (such as in Figure 3)?

---

## [Author Response]

[…] Essential revisions:Two major concerns arose in consultation:1) The protocol for contrast adaptation looks different from the classical ones in the field, and reviewer #1, point 1 requests clarification and a new experiment that should be doable in a short time.2) We would like to see a quantification of how filter shape depends on stimulus variance, and reviewer #2, point 1 provides details of the analyses needed.

We thank the reviewers for their positive assessment of our work and their valuable suggestions. Our point-by-point responses follow.

Reviewer #1:1) If I understand it correctly, the change in stimulus variance is related to a change in correlation time of the light intensity, rather than in its variance about a mean, in order to get the derivatives to scale as intended. Does this cause any problems? What if you just do it the naïve way, by scaling the entire light intensity trace contrast, so that the derivatives and deviations both scale up? Is the answer the same? I think this is potentially important because contrast adaptation is crucially dependent on the timescale on which one computes contrast, and by changing the timescale of the stimulus correlations, one might change regimes. (As an extreme case, if fluctuations were made incredibly slow, one would presumably begin to measure adaptation to the mean, rather than to the contrast.)

In all the experiments presented in the initial submission of our paper, the light intensity obeys a random walk. In a random walk without boundary conditions, the correlation time is infinite, because the average (and most probable) future location is always the current location.P[x(t+τ)|x(t)]=14πDt−−(x(t+τ)−x(t))24DtE[x(t+τ)|x(t)]=x(t) independent of τ

Our led output levels are bounded below by 0 and above by a maximum output level, so the intensity obeys the statistics of a random walk with reflecting boundary conditions. In this case, there is a finite correlation time, which depends on the diffusion constant and the boundary conditions. In the high variance condition, the correlation time is ~12 seconds and in the low variance condition the correlation time is approximately 104 seconds (Figure 4—figure supplement 1D).

We observe adaptation to high variance almost immediately after the switch, while adaptation to low variance occurs over about 10 seconds. Both of these are much faster than the correlation time of the stimulus in their respective conditions.

We initially chose to use a stimulus with uncorrelated derivatives rather than uncorrelated values because previous work has shown that the derivative of the stimulus is more salient to larva than its value, and also to align with our previous work. With uncorrelated derivatives, the light intensities themselves are the same in high and low variance conditions – both span the entire range from 0 to max intensity. In an uncorrelated value stimulus, the high variance condition necessarily includes higher light intensities than are sampled in the low variance condition (compare Figure 4—figure supplement 1C and J). As a result, we worried that adaptation to high variance might be interpreted as due to effects associated with high light intensities, e.g. saturation of receptors or ion channels.

The reviewer’s questions about the structure of our stimulus are well thought out and likely quite common, and we welcome the opportunity to address them in depth in this response letter. We also now include a short discussion and a new Figure 4—figure supplement 1, which shows the stimulus and help visualize what happens when the variance changes.

As the reviewer requested, we also carried out an experiment in which light levels were randomly selected from a normal distribution with constant mean and changing variance. We analyzed these in the LNP framework with the kernel computed from the “turn-triggered-average” stimulus value, and we recapitulated the results from the uncorrelated derivative experiments: larvae adapt to variance via a rescaling of the nonlinear function, and they adapt more quickly to increases of variance than decreases. These results are shown in new Figure 4.

2) The authors frame the adaptation as rescaling the nonlinearity, but what if adaptation rescales the linear filter amplitude. These are mathematically equivalent, and if you do the rescaling of the linear filter amplitude, then the Figure 8B model seems like it could be reconciled with the independent adaptation of the two channels. Is this a problem for excluding Figure 8B for this reason?

We agree that rescaling the kernel and rescaling the input to the nonlinear function are completely equivalent. We also agree that formally, the Figure 8B model works if you are able to rescale the kernel while preserving the linearity of its output. While this is mathematically simple, it is difficult to understand how this might be achieved in a biological system.

Measuring the variance of a signal requires a nonlinear computation, and both spiking and synaptic transmission rectify, so it is natural that biophysical models of adaptation take advantage of these nonlinearities. For instance, Kim and Rieke, (2003) show how an apparent linear rescaling of the filter results from biophysical nonlinearities, including spiking. The LNK model (Ozuysal and Baccus, 2012) accomplishes variance adaptation using a kinetics block following a rectifying nonlinearity. In both of these cases, the LN cascade takes place within a single neuron. Similarly, the Somponlinsky model (Borst et al., 2005) requires a saturating nonlinearity. We are not aware of any mechanistic model of variance adaptation that does not require a nonlinear transformation of the input.

Our finding (Gepner, Mihovilovic and Skanata, 2015) that larvae linearly combine odor and light signals (Figure 8B, this work) requires that these signals be transmitted linearly from the sensory peripheries to some part of the brain responsible for multisensory combination. Because it is difficult to send both positive and negative signals through the same synaptic pathway, this was already somewhat puzzling. But one might, for instance, imagine that for both light and odor, 0 change was represented by a significant amount of excitation which was then balanced by tonic inhibition at the point of combination.

Now we have added the finding that adaptation to variance is upstream of multisensory combination. While one can imagine more elaborate arrangements that would allow the framework of Figure 8B to accommodate variance adaptations, we do not know how variance adaptation might be achieved while preserving linearity. And because variance measurement requires a nonlinear computation, the most parsimonious explanation is that feedback from the rectifying nonlinearity already present in our model should be used to achieve adaptation.

In summary, there is nothing mathematically wrong with this extension of the Figure 8b model to accommodate variance adaptation. But we believe that rescaling the light and odor kernels in response to changes in variance, linearly combining the odor and light filter outputs, and then using that linear combination as the input to a rectifying nonlinearity is biologically implausible. Hence, we sought an alternate model that was more biologically plausible while remaining at least as accurate mathematically.

We have rewritten the discussion of the combination models (which has been moved to the Results section, per reviewer 2’s suggestion) to clarify these points.

3) The multiplicative model is different from additive in the case of polynomial terms of order >1 in the exponential. However, if you expand everything in Taylor series, is the multiplication just allowing a few more higher order interactions between the terms? The relative coefficients of those higher order terms is restricted by this multiplication step. But what happens if you fit models that just allow up-to-cubic interactions between the x_O_ and x_L_ filtered terms before applying the exponential? (This would be a nonlinear interaction before applying the second nonlinearity.) Can you do better than multiplication? If so, would that imply that there's some kind of nonlinear summation of the terms with potentially small nonlinearities that is in fact the best fit? Or is multiplicative really a better model because it requires fewer fitting terms? A BIC evaluation seems like it could resolve this.

We believe (see discussion above) that the multiplicative model (Figure 8C), is more biologically plausible than the linear combination model (Figure 8B) in the face of our finding that larvae adapt to the variance of each sense individually (Figure 7). For consistency, we felt it was also important to show that the multiplicative model was at least as explanatory of our previous experiments as the linear combination model.

When we reexamined the data from Gepner, Mihovilovic and Skanata et al. using the multiplicative combination model, we found that it increased the likelihood of that data given the best-fit model by a statistically significant amount compared to the best fit linear combination model. However, if one looks at the actual predictions made by these two models (Figure 8—figure supplement 1B, Figure 8—figure supplement 2), there are only minute differences. It would be possible, especially with more data, to compare these two models at higher polynomial orders, but presumably the best-fit rate functions would remain quite similar. Our goal was to show that the multiplicative model could explain our past data; we believe the current analysis satisfies this goal and hence have not extended the analysis to higher orders.

Reviewer #2:1) A major point of confusion for me was that the authors claim that the linear filters are conserved across different values of stimulus variance. […] Hence, I would like to see a quantification of how filter shape depends on stimulus variance (e.g., by plotting f_hi vs f_low – if filter shape is preserved across variances, all points should be close to the diagonal – it looks as if filter values close to the peak will lie on the diagonal, but filter values around -6 to -4 seconds prior to the turn should lie off diagonal). Using the actual filter shape for each variance, rather than the filter derived from pooled data, might affect (in both directions) the reported effects of variance on the shape of the nonlinearity.

We have added bootstrapped error bars to the recovered turn triggered averages. We have also re-analyzed the data of Figure 1 using different filter shapes for the low and high variance conditions (Figure 1—figure supplement 2). The derived rate functions are identical within the precision of our measurement. Specifically, using a variance-specific kernel would not change the rate-function presented in Figure 1E.

We estimate the time-varying rescaling parameter(s) (Figure 3, Figure 4, Figure 5 and Figure 7) with maximum likelihood estimation on the static rate function parameters and temporal sequence of rescaling parameter(s). To perform this estimate, we use the pre-computed convolved stimulus as the input to the nonlinear function. It would be computationally challenging to extend our approach to include a time-varying filter on the raw stimulus input. We would not want to use pre-computed low and high variance kernels separately on the low and high variance portions of the stimulus, because an important feature of our analysis is that rescaling emerges from fitting the observed turn rate to the input stimulus without any pre-computation of the variance.

Figure—figure supplement 2 now shows an analysis of Figure 1 data using separate high and low variance kernels. We have rewritten subsection “Larvae Adapt Their Turn-Rate to the Variance in Visual and Olfactory Sensory Inputs” to remove the claim that the linear filter is the same at high and low variance and instead to emphasize that the most dramatic effect is the rescaling of the nonlinear function.

2) There were also some issues with the writing. For example, the abstract is lacking a description of the niche this work aims to fill, and a statement on the general relevance of the findings – as another example, the authors don't example why it is interesting that multisensory integration involves a multiplicative step and what the implication is for behavior. I also often had to go back to their previous paper (Gepner et al., 2015) and dig into the details of their model or look up details on the coordinate transform (see comment below). The rationale for each plot is not made sufficiently clear to the reader. It would help if the authors show more raw data to provide an intuition for the derived plots – e.g., what happens to animal turning following the switch in variance from low to high or high to low (such as in Figure 3)?

We have edited the abstract and introduction to improve readability. We added Figure 1—figure supplement 1 to sketch how we extract high and low variance turn rates from the applied stimulus and observed behavioral raster.